# Human cytomegalovirus evades antibody-mediated immunity through endoplasmic reticulum-associated degradation of the FcRn receptor

Xiaoyang Liu[1], Senthilkumar Palaniyandi[1], Iowis Zhu[1], Jin Tang[1], Weizhong Li[1], Xiaoling Wu[1], Susan Park Ochsner[1], C. David Pauza[2,5], Jeffrey I. Cohen[3] & Xiaoping Zhu [1,4]

Human cytomegalovirus (HCMV) can persistently infect humans, but how HCMV avoids humoral immunity is not clear. The neonatal Fc receptor (FcRn) controls IgG transport from the mother to the fetus and prolongs IgG half-life. Here we show that US11 inhibits the assembly of FcRn with $\beta_2$m and retains FcRn in the endoplasmic reticulum (ER), consequently blocking FcRn trafficking to the endosome. Furthermore, US11 recruits the ubiquitin enzymes Derlin-1, TMEM129 and UbE2J2 to engage FcRn, consequently initiating the dislocation of FcRn from the ER to the cytosol and facilitating its degradation. Importantly, US11 inhibits IgG-FcRn binding, resulting in a reduction of IgG transcytosis across intestinal or placental epithelial cells and IgG degradation in endothelial cells. Hence, these results identify the mechanism by which HCMV infection exploits an ER-associated degradation pathway through US11 to disable FcRn functions. These results have implications for vaccine development and immune surveillance.

---

[1] Division of Immunology, Virginia-Maryland College of Veterinary Medicine, University of Maryland, College Park, MD 20742, USA. [2] Institute of Human Virology, University of Maryland School of Medicine, Baltimore, MD 21201, USA. [3] Laboratory of Infectious Diseases, National Institute of Allergy and Infectious Diseases, NIH, Bethesda, MD 20814, USA. [4] Maryland Pathogen Research Institute, University of Maryland, College Park, MD 20742, USA. [5] Present address: American Gene Technologies, Rockville, MD 20850, USA. Correspondence and requests for materials should be addressed to X.Z. (email: xzhu1@umd.edu)

Human cytomegalovirus (HCMV) is a herpesvirus that infects humans. While most infections with HCMV are asymptomatic, the virus can cause infectious mononucleosis. In either case the virus progresses to latency and infected persons become lifelong infectious carriers. However, HCMV infections pose a life-threatening risk in immunocompromised patients, such as transplant recipients and patients with uncontrolled HIV infection. In addition, due to its ability to infect the developing fetus in utero via placental transmission, HCMV is the leading infectious cause of congenital abnormalities worldwide[1].

HCMV can successfully evade the immune system and establish lifelong latency and persistent virus shedding. Viral infections are normally controlled through antibody-mediated and cell-mediated immunity; the latter involves CD4[+] and CD8[+] T lymphocytes and natural killer (NK) cells. Cellular immunity is essential for limiting HCMV disease[2], and individuals with genetic defects affecting cellular immunity are highly susceptible to HCMV[3,4]. HCMV expresses the US2, US3, US6, US10, US11 proteins that inhibit CD4[+] or CD8[+] T-cell activation through distinct mechanisms[5–14]. The HCMV proteins UL16, UL18, UL40, UL140, UL141, UL142, UL148A, US18, and US20 plus microRNA-UL112 inhibit NK cell activation by moderating MICA, ULBP, and MICB, HLA-E, CD112, CD155- ligands, which normally engage the stimulatory NK cell receptors[15–23]. By altering surface levels of T-cell and NK cell receptor ligands, HCMV interferes broadly with cell-mediated immunity.

Humoral immunity is also important for suppressing HCMV infection. Anti-virus antibodies neutralize virions and stimulate immune cells expressing one or more FcγRs[24]. Several reports noted the importance of antibodies for controlling infection[25]; CMV immunoglobulin is licensed for prophylaxis in transplant recipients[26] and is under study to reduce congenital CMV disease[27]. However, latent HCMV can reactivate and is shed, even in the presence of HCMV-specific IgG[28]. HCMV can circumvent neutralizing antibodies (nAb) because the heavily glycosylated glycoprotein N is poorly recognized[29] or the IgG Fc is found in the viral envelope where it increases the infection of FcγR-expressing cells[30]. Interestingly, the HCMV genome also encodes several decoy FcγRs, which may indirectly prevent the Fcγ-mediated effector consequences of anti-HCMV IgG antibodies[31–33].

The neonatal Fc receptor (FcRn) is composed of a heavy chain (HC) in non-covalent association with $\beta_2$m[34,35]. This association is required for FcRn/$\beta_2$m exit from the endoplasmic reticulum (ER)[36]. Although FcRn shares structural characteristics with MHC class I, it does not present antigenic peptides to cognate T cells due to its narrowed antigen-binding groove[37]. Instead, FcRn binds IgG antibodies in a pH-dependent manner with FcRn binding to the Fc-region of IgG at a pH below 6.5 and releasing IgG at higher pHs[38]. The FcRn is normally transported to early endosomes and has limited cell surface expression. Within these acidic endosomes, FcRn binds endocytosed IgG[39]. Depending on the cell type, FcRn either recycles IgG back to its original cell surface, as is the case with endothelial cells, or transports IgG to the opposite cell surface as is the case with certain polarized epithelial cells in the intestine or placenta. The near neutral pH of the extracellular environment triggers the release of IgG from FcRn. Endocytosed IgG that does not bind FcRn moves to lysosomes where it is degraded[39]. FcRn therefore prolongs the half-life of IgG. As an antibody transporter, FcRn helps to establish passive immunity by carrying maternal IgG across the placental syncytiotrophoblast monolayer, as well as across polarized epithelium lining the respiratory, intestinal, and genital tracts[40–42]. Also, FcRn is a target for delivering drugs, therapeutics, and vaccines[43–45]. In all, FcRn plays a critical role in establishing early neonatal immunity and is involved with immune responses to both natural infection and vaccination.

Little is currently known about the interaction between HCMV and FcRn. HCMV infects placental trophoblasts, epithelial cells, endothelial cells, and hematopoietic stem cells[46–50]; Among the hematopoietic cell lineage, FcRn expression is restricted to myeloid cells, including macrophages and dendritic cells[39,43,51]. Maternal immunity is central to protection of the fetus because infection can occur when neutralizing IgG is low[48], although the role of FcRn has remained somewhat elusive[52]. As FcRn is important in passive immunity, its inactivation could lead to superinfection of an unprotected developing fetus. Here, we have identified that the HCMV glycoprotein US11 specifically captures human FcRn, inhibits its antibody trafficking functions, and causes its degradation in a process known as endoplasmic reticulum-associated degradation (ERAD). This process may be involved with dampening mucosal and maternal immunity and reducing IgG half-life in blood and tissues. We therefore propose a mechanism through which HCMV escapes antibody-mediated immunity.

## Results

**HCMV glycoprotein US11 interacts with FcRn.** Genes encoding HCMV proteins, US2, US3, US6, US10, US11, UL16, and UL18 were cloned by PCR amplification of viral DNA. We then probed for interactions between FcRn and each individual HCMV protein. HeLa[FcRn] cells were transfected with plasmids encoding each of the HA-tagged HCMV cDNAs. Owing to its affecting FcRn stability, we characterized US11 and FcRn interaction in this study.

HeLa[FcRn] or HeLa cells were transfected with plasmids encoding HA-tagged US11 cDNA. Cell lysates were used for immunoprecipitation with either anti-FLAG (for FcRn) (Fig. 1a) or anti-HA (for US11) (Fig. 1b) mAb. The anti-FLAG Ab co-immunoprecipitated US11 protein (Fig. 1a) and the anti-HA Ab co-immunoprecipitated FcRn HC (Fig. 1b). We analyzed the co-localization between FcRn and US11 using confocal microscopy (Fig. 1c). FcRn appeared in a punctate pattern within HeLa[FcRn] cells, while US11 was highly colocalized with FcRn in HeLa[FcRn+US11] cells (Fig. 1c). To identify the specificity, we co-expressed FcRn with US2, and US11 with HFE that has an MHC class I-like structure in HeLa cells. We failed to detect an interaction between US11 and HEF (Fig. 1d, e), US11 and endogenous transferrin receptor (TfR1) (Supplementary Fig. 1a), or FcRn and US2 (Supplementary Fig. 1b, c) in a reciprocal immunoprecipitation.

To further characterize the interaction between FcRn and US11, we infected several types of cells, including primary human umbilical vein endothelial cells (HUVEC), endothelial HMEC-1, THP-1 cells, and intestinal epithelial Caco-2, with clinical strain HCMV (MOI 5). The infection was verified by detecting HCMV phosphoprotein 65 (PP65) (Supplementary Fig. 2). At day 2 post infection (p.i.), we determined that anti-US11 Ab co-immunoprecipitated with FcRn HC in infected HUVEC cells (Fig. 1f). Similarly, anti-FcRn Ab was also found to co-immunoprecipitate with US11 protein in infected HUVEC cells (Fig. 1g). The US11 and FcRn interaction was also present in either monocytic THP-1 (Supplementary Fig. 3a, b) or PMA-differentiated macrophage-like THP-1 cells (Supplementary Fig. 3c, d), HMEC-1 (Supplementary Fig. 3e, f), and Caco-2 (Supplementary Fig. 3g, h). These results indicate that that FcRn and US11 interaction occurs in multiple cell types during HCMV infection. Using GST-tagged US11 or cytoplasmic tail (CT) FcRn proteins, we also found that the main site of contact between US11 and FcRn is between their extracellular domains (Supplementary Fig. 4a–c).

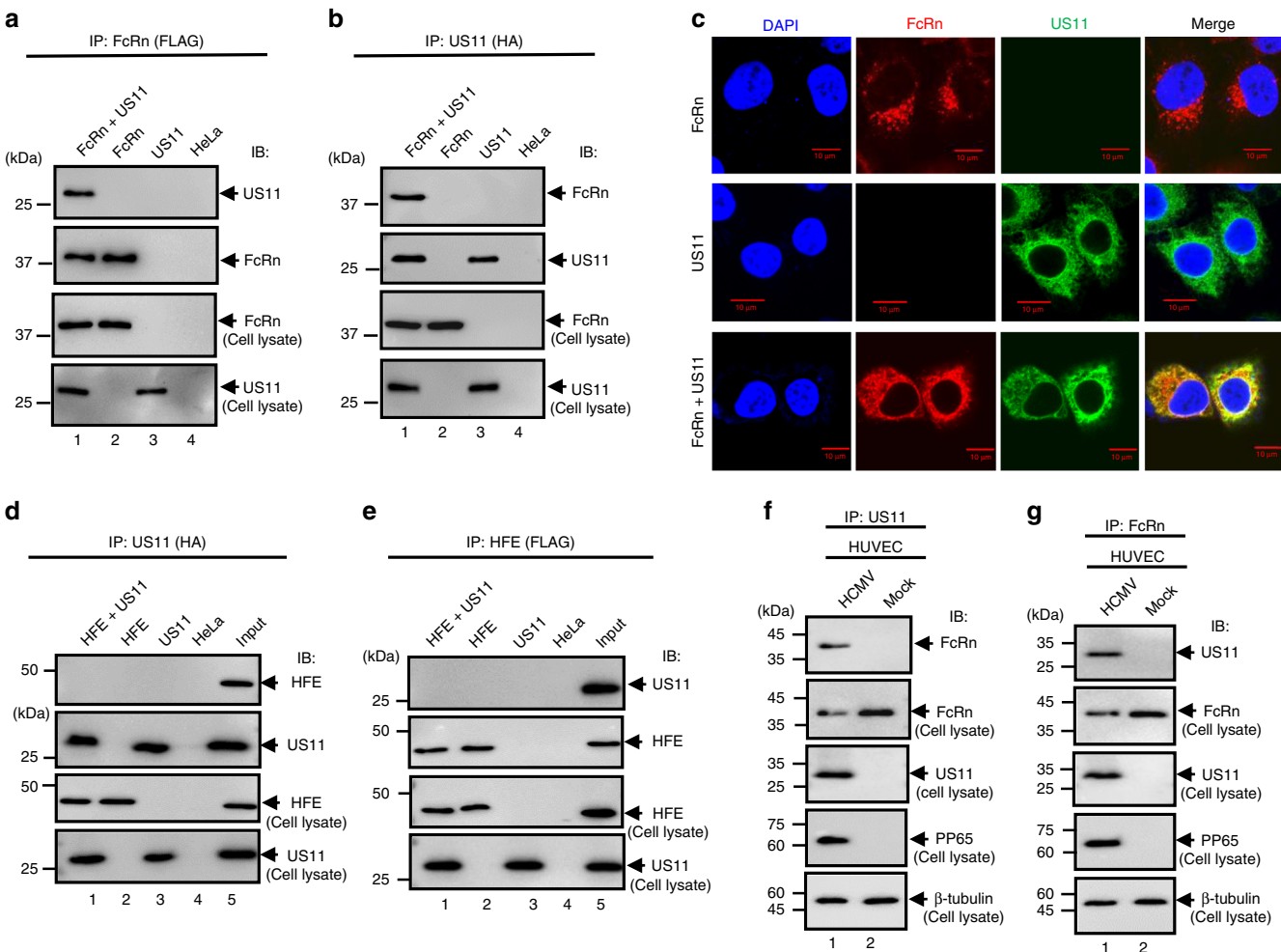

**Fig. 1** FcRn interacts with HCMV US11. **a, b** The cell lysates from HeLa$^{FcRn+US11}$ (lane 1), HeLa$^{FcRn}$ (lane 2), HeLa$^{US11}$ (lane 3), and HeLa control cells (lane 4) were immunoprecipitated by mAb anti-HA for US11 or anti-FLAG for FcRn. The immunoprecipitates were subjected to western blotting with anti-FLAG or HA mAb as indicated. Cell lysate from each sample with equal amounts of total protein (input, 20 μg) were blotted with the indicated Abs. **c** Co-localization of FcRn and US11 in HeLa$^{FcRn+US11}$ cells. HeLa$^{FcRn}$ cells or HeLa$^{US11}$ cells were used as a control. Puncta that appear yellow in the merged images (right panel) indicate co-localization of FcRn with US11 protein. The nuclei were stained with DAPI (blue). Scale bar: 10 μm. **d, e** Cell lysates from HeLa$^{HFE+US11}$ (lane 1), HeLa$^{HFE}$ (lane 2), HeLa$^{US11}$ (lane 3), and HeLa control cells (lane 4) were immunoprecipitated with mAb anti-HA for US11 or anti-FLAG for HFE, respectively. The immunoprecipitates were subjected to western blotting with anti-FLAG or HA mAb as indicated. The cell lysates (input) were blotted as control. **f, g** US11 interacts with FcRn in HCMV-infected human primary umbilical vein endothelial cells (HUVEC). HUVEC were infected with HCMV at an MOI of 5. At day 2 p.i., the cell lysates from infected or mock-infected HUVEC were immunoprecipitated with anti-US11 Ab (**f**) or anti-FcRn Ab (**g**). The immunoprecipitates were subjected to 12% SDS-PAGE electrophoresis under reducing conditions, then transferred to a nitrocellulose membrane for western blotting with anti-FcRn or US11 Ab as indicated. The cell lysates (20 μg) were blotted as controls

**US11 expression reduces FcRn traffic to the early endosome**. In most cell types, most FcRn resides in acidic endosomes where FcRn binds pinocytosed or endocytosed IgG[39]. We hypothesized that FcRn distribution in the endosome would be affected by the interaction of US11 with FcRn HC. In HeLa$^{FcRn+US11}$ cells, co-localization of FcRn and the early endosomal marker EEA1 (early endosomal Ag-1) was significantly decreased compared with control HeLa$^{FcRn}$ cells (Fig. 2a, b). This suggests that US11 impairs FcRn endosomal trafficking. The reduction of co-localization between FcRn and EEA1 is unlikely due to the decreased expression of FcRn in HeLa$^{FcRn+US11}$ cells because the same HeLa$^{FcRn}$ cell line was used for transfecting the US11 expression plasmid and for control cells to monitor the level of FcRn expression. Although the over-expression of FcRn does not affect its intracellular trafficking pattern[53,54], the over-expression of US11 may cause extensive remodeling of intracellular membranes. To investigate this possibility, we compared the

co-localization of transferrin receptor (TfR1, CD71) with EEA1 between HeLa$^{FcRn}$ and HeLa$^{FcRn+US11}$ cells (Fig. 2c). We found there was no significant difference in co-localization of TfR1 with EEA1 between HeLa$^{FcRn}$ and HeLa$^{FcRn+US11}$ cells (Fig. 2d), suggesting that the potential remodeling of cellular membranes by the over-expression of US11 may not affect endogenous protein trafficking. Altogether, these data confirm that the routing of FcRn to endosomes in epithelial cells is significantly reduced by US11.

The functional FcRn molecule consists of the HC bound to $\beta_2$m, we tested whether US11 interacts with FcRn HC or FcRn-$\beta_2$m. Lysates from HeLa$^{FcRn+US11}$ were immunoprecipitated with anti-HA (for US11) or anti-$\beta_2$m Ab and blotted with anti-$\beta_2$m (BBM1) Ab to detect $\beta$2m (Fig. 2e) or anti-HA Ab to detect US11 (Fig. 2f). Immunoprecipitates were sequentially blotted with Abs against US11, FcRn, or $\beta_2$m. Anti-HA Ab failed to co-immunoprecipitate $\beta_2$m (Fig. 2e). Similarly, $\beta_2$m mAb did not

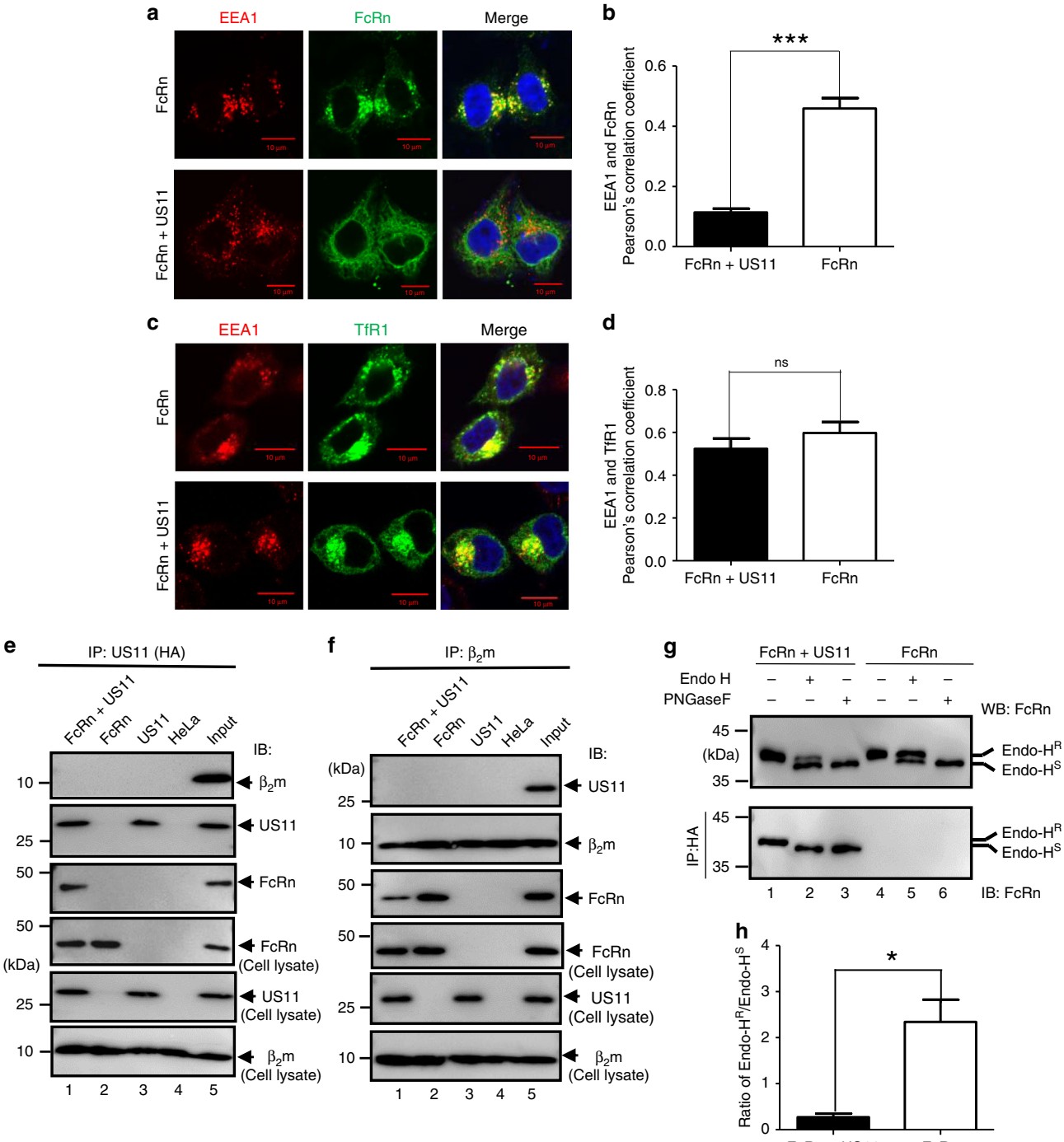

pull down US11 (Fig. 2f). However, an Ab against either HA or $\beta_2$m co-immunoprecipitated FcRn HC. These data strongly suggest that HCMV US11 interacts with $\beta_2$m-free FcRn HC.

FcRn has one *N*-linked glycosylation site[55]. To identify the glycosylation status of FcRn in US11+ cells, we treated cell lysates with either Endo H glycosidase, which cleaves high-mannose oligosaccharides formed only in the ER, or with PNGase amidase, which cleaves hybrid and complex oligosaccharides formed in both the ER and golgi complex. FcRn HC from HeLa^FcRn+US11 cells was much more sensitive to Endo H digestion than FcRn from HeLa^FcRn cells (Fig. 2g, top panel). As expected, FcRn from both cell lines was sensitive to PNGase F digestion. We then tested the Endo H sensitivity of FcRn in anti-US11 immunoprecipitates from HeLa^FcRn+US11 cells. As shown in Fig. 2g (lower

panel), FcRn HC (lane 2) in a US11 immunoprecipitation from HeLa^FcRn+US11 cells demonstrated a mobility similar to FcRn HC after PNGase F digestion (lane 3). The full sensitivity of FcRn HC to Endo H digestion conforms to an ER-specific glycosylation pattern of FcRn within the FcRn/US11 complex (Fig. 2h). Overall, these findings strongly support our conclusion that the known failure of newly synthesized FcRn to assemble with $\beta_2$m and traffic to the early endosome during HCMV infection is due to its physical association with US11.

### US11 is necessary for FcRn degradation during HCMV infection.
To examine how US11 affects FcRn expression, we first compared the expression levels of surface and intracellular FcRn in HeLa^FcRn or HeLa^FcRn+US11 cells using flow cytometry. We

**Fig. 2** US11 expression retains FcRn in the endoplasmic reticulum (ER). **a**, **c** FcRn or CD71 appearance in the endosome in HeLa$^{FcRn+US11}$ and HeLa$^{FcRn}$ cells. HeLa$^{FcRn+US11}$ and HeLa$^{FcRn}$ cells were transfected with a plasmid expressing human CD71-GFP (green). Both cells were immunostained for FcRn (green) and EEA1 (red). **b**, **d** Averages of the EEA1 and FcRn or CD71 co-localization in HeLa$^{FcRn+US11}$ and HeLa$^{FcRn}$ cells. Pearson's correlation coefficient was measured in ten different optical regions (100 cells total) in each experiment. Error bars represented mean ± SEM of ten different optical regions. The nuclei were stained with DAPI (blue); co-localization of two molecules appears in yellow. Scale bar: 10 μm. Similar images were seen from at least three independent staining experiments. EEA1: early endosome antigen 1. ***$P < 0.001$, NS: no significance by Student's $t$-test. **e**, **f** $\beta_2$m or US11 does not co-immunoprecipitate with US11 or $\beta_2$m protein. Cell lysates from HeLa$^{FcRn+US11}$ (lane 1), HeLa$^{FcRn}$ (lane 2), HeLa$^{US11}$ (lane 3), and HeLa cells (lane 4) were immunoprecipitated by anti-HA mAb (**e**), anti-$\beta_2$m Ab (**f**). All immunoprecipitates or cell lysates (20 μg) as internal controls were subject to western blotting with corresponding Abs as indicated. **g**, **h** Sensitivity of US11-associated FcRn HC to Endo-H digestion. **g** Native FcRn in cell lysates (top panel) or proteins immunoprecipitated by HA mAb (bottom panel) were digested by mock (lanes 1 and 4), Endo-H (lanes 2 and 5), or PNGase F (lanes 3 and 6) for 2 h at 37 °C, respectively. All immunoprecipitates or cell lysates (20 μg) were subject to western blotting with corresponding Abs as indicated. The ratio of Endo H-resistant (Endo-H$^R$) FcRn HC to Endo H-sensitive (Endo-H$^S$) FcRn HC from HeLa$^{FcRn+US11}$ and HeLa$^{FcRn}$ cells were compared by the ratio of the band density of glycosylated FcRn to that of the deglycosylated FcRn (**h**). The band density of Endo-H-sensitive or resistant FcRn (**g**, top panel, lanes 2 and 5) was quantified by the software Image Lab 5.2. *$P < 0.05$ by Student's $t$-test. Error bars represented mean ± SEM of three independent repeats. R: Resistant; S: Sensitive

found that US11 diminished both the surface and intracellular expression level of FcRn (Fig. 3a). In contrast, US11 affected neither the surface nor the intracellular expression level of HFE under the same conditions (Fig. 3b), suggesting a specific downregulation of FcRn by US11.

To specifically monitor the rate of FcRn HC degradation, we performed a quantitative cycloheximide (CHX) chase assay. HeLa$^{FcRn+US11}$ (Fig. 3c) and HeLa$^{FcRn}$ (Fig. 3e) cells were treated with CHX (100 μg/ml) and FcRn intensity detected in western blot was measured by the Image Lab for the indicated times. In HeLa$^{FcRn+US11}$ cells, the US11 expression induced a significant and time-dependent decrease in FcRn expression (Fig. 3c, d) in comparison with that of HeLa$^{FcRn}$ cells (Fig. 3e). FcRn in US11$^+$ cells also had a shorter half-life, compared to the long-term stability of FcRn in control cells. Therefore, US11 promotes FcRn protein degradation. In contrast, we did not detect a significant change in the levels of β2m (Fig. 3f), suggesting that the effect of US11 on FcRn levels was not due to either US11- or CHX-induced cytotoxicity.

To identify whether this effect of US11 on endogenous FcRn is seen during HCMV infection, we infected HUVEC or Caco-2[40]. Forty-eight hours post infection, we performed a quantitative CHX-chase analysis. Compared to mock-infected controls (Fig. 3i and Supplementary Fig. 5c), the expression levels of FcRn HC were significantly decreased in HCMV-infected HUVEC cells (Fig. 3g, h) or Caco-2 cells (Supplementary Fig. 5a, b). FcRn mRNA levels did not change in Caco-2 cells (Supplementary Fig. 5f–i), indicating that FcRn is not regulated by HCMV at the transcriptional level. This FcRn downregulation during infection was further supported by decreasing intracellular levels of FcRn in HCMV-infected THP-1 and HMEC-1 cells using flow cytometry (Supplementary Fig. 6a, b). To identify the role of US11 in this process, we used two independent US11 small-interfering RNA (siRNA) species to knock-down US11 in HCMV-infected cells. FcRn degradation was significantly reduced in US11 siRNA-treated HUVEC (Fig. 3h, k) or Caco-2 (Supplementary Fig. 5b, e), although β2m levels were unaffected in HUVEC cells (Fig. 3j, k) or Caco-2 cells (Supplementary Fig. 5d, e). We also noticed that the FcRn level at 240 min post chase (Fig. 3k and Supplementary Fig. 5e) was moderately restored by US11 siRNA in virus infected cells in comparison with mock-infected cells (Fig. 3i and Supplementary Fig. 5c). It is likely that this result was associated with the incomplete blocking of US11 expression by US11 siRNA, which was shown in a US11 blot (Fig. 3k and Supplementary Fig. 5e, middle panels). In addition, we also found a mutant HCMV ΔUS2-12 that lacks the US2-12 genes failed to induce a significant and time-dependent decrease in FcRn expression in HCMV ΔUS2-12-infected HEE$^{FcRn}$ cells in comparison with that

of mock or HCMV-infected cells (Supplementary Fig. 7). Taken together, these results strongly suggest that US11 is required for decreasing the intracellular concentration of FcRn during HCMV infection.

**US11 recruits Derlin-1 and TMEM-129 proteins to engage FcRn**. Derlin-1 can interact with US11[56,57] and facilitates movement of misfolded proteins through the ER membrane[58]. We identified Derlin-1 as a potential partner of the FcRn-US11 complex. The US11–Derlin-1 interaction is dependent on a polar glutamine residue (Q192) in the US11 transmembrane domain[59]. To identify interactions between FcRn, US11, and Derlin-1, we transfected HeLa$^{FcRn}$ cells with plasmids encoding Derlin-1 and either a wild-type or mutant US11 Q192L. Using an anti-FLAG Ab against FcRn, we co-immunoprecipitated US11 and Derlin-1 (Fig. 4a, lane 1), or mutant US11 without Derlin-1 (Fig. 4a, lane 2). Using an anti-Myc Ab against Derlin-1, we co-immunoprecipitated both FcRn HC and US11 (Fig. 4b, lane 1), but failed to pull down FcRn in the presence of mutant US11 Q192L (Fig. 4b, lane 2), verifying that Derlin-1 is incapable of binding mutant US11 Q192L. Furthermore, anti-FLAG Ab against FcRn did not co-immunoprecipitate Derlin-1 in the absence of US11 (Fig. 4c, lane 1). Conversely, the FcRn levels were maintained in HeLa$^{FcRn+US11\ Q192L}$ cells during the CHX treatment, suggesting that the observed FcRn decrease is mediated through interactions between US11 and Derlin-1 (Fig. 4d, e). Altogether, these data suggest that the Derlin-1 binding activity of US11 is required for FcRn degradation.

Derlin-1 is known to interact with the E3 ligases. A thorough screening of the candidate ligases in the presence of both US11 and Derlin-1 showed that the E3 ligase TMEM129[60,61] was recruited to the FcRn/US11/Derlin-1 complex (Fig. 4f, g). In the absence of US11 expression, immunoprecipitation of FcRn failed to pull down TMEM129 (Fig. 4f, lane 2). In addition, TMEM129 was capable of directly pulling down Derlin-1 in HeLa$^{FcRn}$ cells without US11 (Fig. 4g, lane 2), suggesting that the binding of TMEM129 to Derlin-1 is US11-independent. The recruitment of TMEM129 to the FcRn/US11 complex is through Derlin-1, as the wild-type US11 co-precipitated with TMEM129 while the Derlin-binding mutant US11 Q192L failed to pull down TMEM129 (Supplementary Fig. 8a).

With this information in hand, we further decided to identify the role of TMEM129 in the downregulation of FcRn in HeLa$^{FcRn+US11}$ cells. We found that depletion of TMEM129 using siRNA (Fig. 4h, bottom) reduced the loss of FcRn expression in HeLa$^{FcRn+US11}$ cells (Fig. 4h, i), suggesting that TMEM129 is critical for US11-mediated degradation of FcRn. We

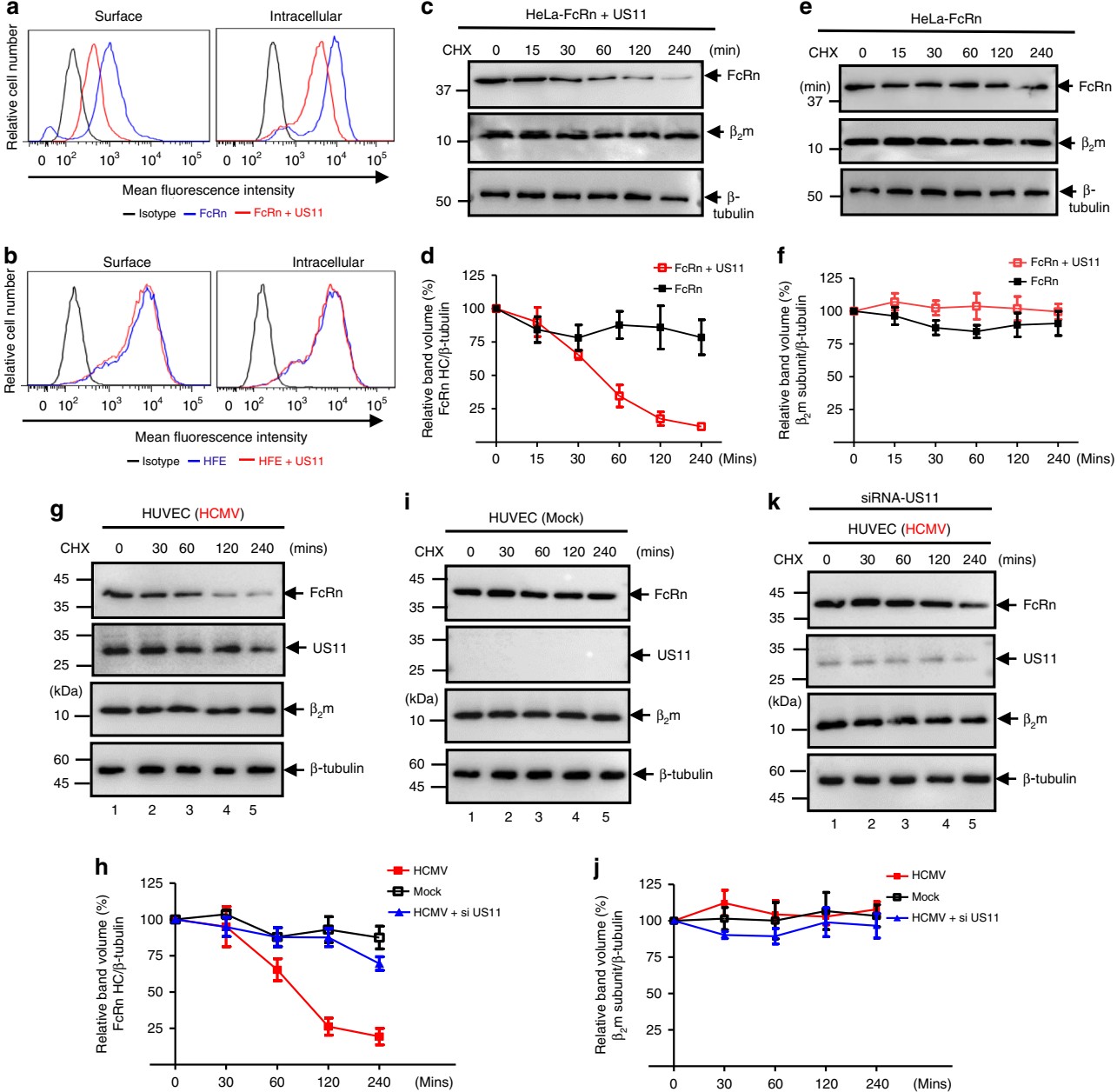

**Fig. 3** US11 protein mediates FcRn degradation. **a**, **b** Cell surface and intracellular expression patterns of FcRn and HFE in either fixed or permeabilized US11-expressing cells were measured by flow cytometry. Cells were stained as described in Methods. The red or blue histograms represent staining of cells with anti-FLAG (FcRn or HFE)-specific Ab with or without expression of US11, and the black histograms represent cells stained with isotype-matched IgG. The staining was performed three times with similar results. The mean fluorescence intensity (MFI) is shown on the x-axis, and the relative cell number on the y-axis. Results are expressed as histograms of fluorescence intensity (log scale). **c–f** HeLa$^{FcRn}$ cells were transfected with US11 plasmids for 24 h. HeLa$^{FcRn+US11}$ (**c**) and HeLa$^{FcRn}$ (**e**) cells were then treated with CHX (100 μg/ml) for the indicated time. The cells were lysed after CHX treatment and cell lysates (20 μg) were subject to western blotting with corresponding Abs as indicated. The level of remaining FcRn (**d**) and $\beta_2$m (**f**) at different time points was quantified as the percentage of the β-tubulin level. **g–k** HUVEC cells were infected with clinic strain HCMV (MOI 5) (**g**) or mock-infected (**i**) for 48 h. The infected cells were also transfected with 20 nM US11 siRNA oligomers (**k**). Forty-eight hours later, cells were then treated with CHX (100 μg/ml) for the indicated time. The cells were lysed after CHX treatment and cell lysates (20 μg) were subject to western blotting with corresponding Abs as indicated. The level of remaining endogenous FcRn (**h**) and $\beta_2$m (**j**) in HUVEC cells at different time points was quantified as the percentage of the β-tubulin level. The percentage of time point 0 (min) is assigned a value of 100% and the values from other time points are normalized to this value (**d**, **f**, **h**, **j**). Error bars represented mean ± SEM of three independent repeats

therefore conclude that TMEM129 is responsible for US11-mediated FcRn degradation.

**FcRn cytoplasmic tail is necessary for degradation by US11**. We have shown that the extracellular domains of US11 and FcRn interact with each other (Supplementary Fig. 4b, c). It is unknown whether the cytoplasmic tail (CT) of FcRn plays a role in US11-mediated degradation. To elucidate the involvement of the FcRn CT, we generated a tailless FcRn (Fig. 5a), leaving five residues allowing the proper insertion of the protein into the membrane[62]. We and others have shown that similarly tailless FcRn behaves

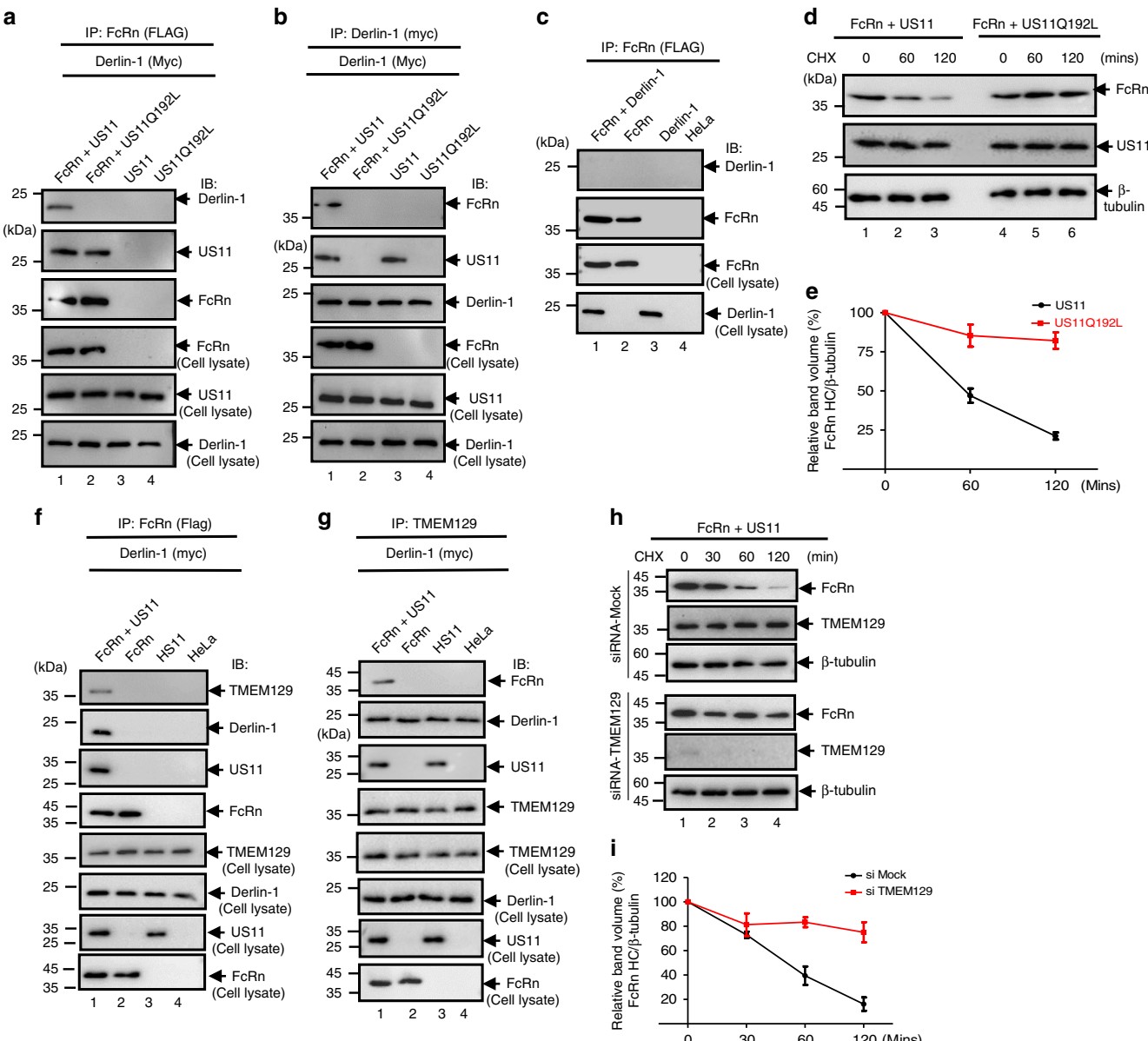

**Fig. 4** US11 recruits FcRn to Derlin-1 and TMEM129 protein complex. **a–c** US11 recruits FcRn to the Derlin-1 complex. US11Q192L represents a mutant US11 in which Q192 is replaced with leucine. Stable HeLa^FcRn, HeLa^FcRn+US11, HeLa^FcRn+US11 Q192L, HeLa^US11, and HeLa^US11 Q192L cell lines were transfected with a plasmid encoding myc-tagged Derlin-1. Forty-eight hours after transfection, the cell lysates (0.5 mg) were immunoprecipitated with mAb anti-FLAG for FcRn (**a**, **c**) or anti-myc for Derlin-1 (**b**). Non-transfected HeLa^FcRn or HeLa cells were used as a negative control. **d, e** FcRn in mutant US11^Q192L transfected cells resists degradation. HeLa^FcRn+US11 and HeLa^FcRn+US11* cells were treated with CHX (100 μg/ml) for the indicated time (**d**). Cells were lysed after CHX treatment and cell lysates (20 μg) were subject to western blotting with corresponding Abs as indicated. The level of remaining FcRn (**e**) at different time points was quantified as the percentage of the β-tubulin level. **f, g** US11 recruits FcRn to TMEM129 complex. HeLa^FcRn+US11 (lane 1), HeLa^FcRn (lane 2), HeLa^US11 (lane 3), and HeLa cells were transfected with Derlin-1 plasmid. Forty-eight hours later, the cell lysates were immunoprecipitated by mAb anti-FLAG for FcRn (**f**) or anti-TMEM129 Ab (**g**). **h, i** TMEM129 is involved in US11-mediated FcRn degradation. The HeLa^FcRn+US11 cells were transfected with 20 nM TMEM129 siRNA oligomers (**h**, bottom). Forty-eight hours later, cells were then treated with CHX (100 μg/ml) for the indicated time. The cells were lysed after CHX treatment and cell lysates (20 μg) were subjected to western blotting with Abs as indicated. The level of remaining FcRn (**i**) in TMEM129 siRNA-treated cells (red) or mock-treated cells (black) at different time points was quantified as the percentage of the β-tubulin level. All immunoprecipitates or cell lysates (20 μg) as controls were subject to western blotting with corresponding Abs as indicated (**a**, **b**, **c**, **f**, and **g**). The percentage of time point 0 (min) is assigned a value of 100% and the values from other time points are normalized to this value (**e**, **i**). Error bars: mean ± SEM of three independent repeats

like full-length FcRn in assembling with β₂m and pH-dependent binding to IgG[53]. To examine this, HeLa^US11 cells were transfected with either a WT FcRn or FcRn-tailless plasmid (Fig. 5b). The fate of both proteins was further examined using a CHX-chase experiment. In the absence of proteasome inhibitor, FcRn HC was degraded by 60 min post chase (Fig. 5b, c). This was in marked contrast with tailless FcRn, which persisted in the presence of US11 (Fig. 5b, c). To identify the critical region within the FcRn CT that is responsible for its degradation, we constructed a series of C-terminal FcRn deletion mutants and examined their susceptibility to US11-mediated degradation. We found that deletion of a single C-terminal alanine residue affected

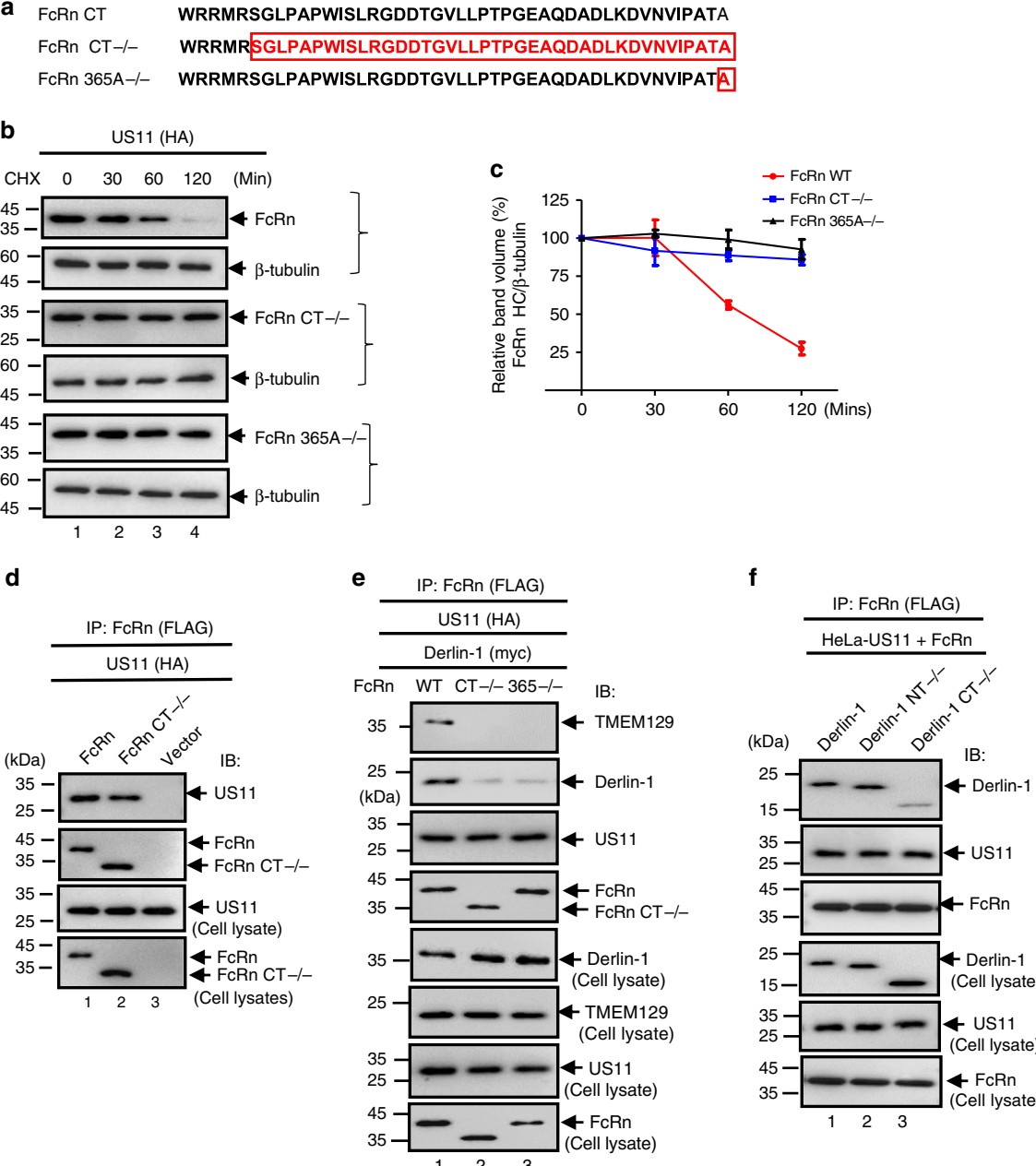

**Fig. 5** The cytoplasmic tail of FcRn contributes to US11-mediated degradation. **a** Depiction of tailless FcRn (CT−/−) and FcRn HC deleting alanine residue 365 (365A−/−) in the cytoplasmic tail. Letter(s) in red represent(s) the deleted amino acid(s). **b, c** Tailless FcRn or FcRn 365A−/− resists degradation in the presence of US11. HeLa$^{US11}$ (**b**, top), HeLa$^{US11+FcRn\ CT−/−}$ (**b**, middle), or HeLa$^{US11+FcRn365−/−}$ cells (**b**, bottom) were treated with CHX (100 μg/ml) and chased for the indicated time in the absence of proteasome inhibitors. The cells were lysed after CHX treatment and cell lysates (20 μg) were subject to western blotting with Abs as indicated. The level of wild-type FcRn (**c**, red), tailless FcRn (**c**, blue), and mutant FcRn 365A−/− (**c**, black) in HeLa$^{US11}$ cells at different time points was quantified as the percentage of the β-tubulin level. The percentage of time point 0 (min) is assigned a value of 100% and the values from other time points are normalized to this value (**c**). Error bars represented mean ± SEM of three independent repeats. **d** The US11 interacts with tailless FcRn protein. The cell lysates from HeLa$^{FcRn+US11}$ (lane 1), HeLa$^{US11+FcRn\ CT−/−}$ (lane 2), and HeLa$^{control}$ (lane 3) were immunoprecipitated by anti-FLAG Ab for FcRn. The US11 molecules that coprecipitate in the complex are indicated. **e** The cytoplasmic tail of FcRn is required for tightly binding to Derlin-1 in the presence of US11. HeLa$^{FcRn+US11}$ (lane 1), HeLa$^{US11+FcRn\ CT−/−}$ (lane 2), and HeLa$^{US11+FcRn\ 365A−/−}$ (lane 3) cells were transfected with Derlin-1 plasmid. Forty-eight hours later, cell lysates were immunoprecipitated with anti-FLAG Ab to detect FcRn. **f** The C-terminus of Derlin-1 is required for tightly binding to FcRn in the presence of US11. The HeLa$^{US11+FcRn}$ stable cells were transfected with a plasmid encoding Myc-tagged Derlin-1 (WT), Derlin-1 lacking its N-terminus (NT−/−) or C-terminus (CT−/−), respectively. Forty-eight hours after transfection, the cell lysates (0.5 mg) were immunoprecipitated with rabbit anti-FLAG Ab. All immunoprecipitates or cell lysates (20 μg) as internal controls were subject to western blotting with corresponding Abs as indicated (**d–f**)

its susceptibility to US11-induced degradation in a manner like that observed after deleting the entire FcRn CT (Fig. 5b, c). Indeed, the half-life of FcRn-365A−/− was comparable with that of FcRn-tailless in HeLa$^{US11}$, while the alanine deletion also rendered FcRn resistant to US11-induced degradation (Fig. 5b, c). We therefore demonstrate that the FcRn CT is necessary for its degradation.

To gain a deeper understanding of how the FcRn CT is involved during US11-induced degradation of FcRn HC, we next determined how FcRn CT interacts with US11 and Derlin-1. Using co-precipitation (Fig. 5d), we found that our FcRn tailless mutant maintained an interaction with US11. However, FcRn tailless had a markedly reduced interaction with Derlin-1 (Fig. 5e; compare lanes 1 and 2). In addition, we found that FcRn-365A−/− also had a similarly decreased interaction with Derlin-1 (Fig. 5e; compare lanes 1 and 3). Finally, immunoprecipitation of tailless FcRn failed to precipitate TMEM129, suggesting that the interaction between FcRn CT and Derlin-1 is also important for TMEM129 recruitment (Fig. 5e). When HeLa$^{US11+FcRn}$ cells were transfected with a plasmid encoding Derlin-1, Derlin-1 lacking its N-terminus (NT−/−) or C-terminus (CT−/−), and FcRn was immunoprecipitated, we found that the C-terminus of the Derlin-1 was required for tight binding to FcRn in the presence of US11 (Fig. 5f, lane 3). Taken together, these results suggest that FcRn CT is required for US11-induced degradation, perhaps via its interaction with Derlin-1.

**US11/Derlin-1/TMEM129/Ube2J2 complex induces FcRn dislocation, ubiquitination, and degradation**. Ubiquitination of a protein substrate is a critical step in protein degradation[63]. We then investigated whether US11 regulates FcRn turnover through a ubiquitination-dependent mechanism. We transfected HeLa$^{FcRn}$ cells with either WT or a mutant US11Q192L. We found that US11 specifically induced FcRn ubiquitination in the presence of proteasome inhibitor, MG132 (Fig. 6a, b). However, the mutant US11 Q192L failed to trigger FcRn ubiquitination (Supplementary Fig. 8b), suggesting that the Derlin-1 is required for FcRn ubiquitination. Furthermore, FcRn tailless (Fig. 6c, lane 2) and FcRn CT365A−/− (Fig. 6c, lane 3) exhibited significantly less ubiquitination in comparison with WT FcRn (Fig. 6c, lane 1), indicating that FcRn CT is necessary for US11-induced FcRn ubiquitination.

We also detected FcRn proteins as both slow and fast migrating bands (Fig. 6c, lane 1, middle). We found that FcRn from HeLa$^{FcRn+US11}$ cells migrated slowly during early CHX-chase time points (Fig. 6d, lane 4) and faster (Fig. 6d, lanes 5–6) during later time points in the presence of both US11 and MG132. We reasoned this faster band might represent removal of the FcRn glycan by cytosolic N-glycanase. To determine this, we subjected HeLa$^{FcRn+US11}$ or HeLa$^{FcRn}$ cells to a subcellular fractionation and used the extracts from either the pellet (membrane) or supernatant (cytosol) for blotting analysis of FcRn. Using Endo H or PNGase F digestion, we found that the slower migrating form of FcRn in the membrane fractions of HeLa$^{FcRn}$ corresponded to Endo H-resistant protein (Fig. 6e, middle). However, FcRn HC from HeLa$^{FcRn+US11}$ cells was much more sensitive to Endo H digestion (Fig. 6e, middle), and some FcRn was present in the membrane fraction only and displayed normal maturation into a slower migrating form. As shown in Fig. 6e (middle), the faster migrating bands (lanes 3 and 4) from the membrane and cytosol fractions of HeLa$^{FcRn+US11}$ cells had a mobility similar to the sensitive bands from Endo H digestion. As expected, FcRn from either source was sensitive to PNGase F digestion (Fig. 6e, bottom). We therefore conclude that the two forms of FcRn represent a glycosylated version, which co-fractionates with the ER membrane, and a cytosolic, non-glycosylated version of FcRn. The origin of this non-glycosylated FcRn form could be due to dislocation of FcRn from the ER; the cytosolic N-glycanase removes the glycan from the FcRn.

The TMEM129 contains an atypical RING domain with intrinsic E3 ubiquitin ligase activity[60,61]. To show that TMEM129 is necessary for FcRn ubiquitination, we knocked down TMEM129 expression (Fig. 7a, middle, lane 2). We then immunoprecipitated FcRn and found the robust FcRn ubiquitination induced by US11 (Fig. 7a, top, lane 1) was abrogated in TMEM129 siRNA-treated cells (Fig. 7a, top, lane 2). The atypical RING-C2 domain of TMEM129 creates a binding platform for E2 conjugating enzymes. Previous studies[60,61] implied that Ube2J1 or Ube2J2 is a potential player in the TMEM129-mediated FcRn ubiquitination, we observed that knock-down of Ube2J2, but not Ube2J1, nearly abolished US11-induced FcRn ubiquitination (Fig. 7b, top, lane 1) and rescued FcRn from degradation in HeLa$^{FcRn+US11}$ cells (Fig. 7c, d). We conclude that the TMEM129 recruits Ube2J2 for US11-induced FcRn ubiquitination, leading to its eventual ER dislocation and proteasome degradation.

**US11 reduces IgG transcytosis in epithelial cell monolayers**. To examine whether HCMV infection or US11 alone affects FcRn-mediated IgG transcytosis across polarized epithelial cells, we first tested whether the association of FcRn with US11 affects FcRn binding to IgG. FcRn is known to bind IgG at pH below 6.5 and to release IgG at neutral pH[38]. Lysates from HeLa cells expressing FcRn and/or US11 were incubated with IgG-Sepharose at either pH 6.0 or pH 7.4. Eluates and cell lysates were subjected to analysis by western blot. As expected, FcRn from HeLa$^{FcRn}$ cells bound IgG at pH 6.0 (Fig. 8a, lane 2), but not at pH 7.4 (Fig. 8b, lane 2). Similarly, $\beta_2$m from the IgG binding eluates was detected at pH 6.0 (Fig. 8a, lane 2), but not at pH 7.4 (Fig. 8b, lane 2). We did not detect US11 proteins in the FcRn binding eluates of IgG beads at pH 6.0 (Fig. 8a, lane 1). Furthermore, FcRn and $\beta_2$m levels (Fig. 8a, lane 1) eluted from IgG beads were remarkably decreased in HeLa$^{FcRn+US11}$ cells when compared to in HeLa$^{FcRn}$ cells alone (Fig. 8a, lane 2). These data strongly suggest that the association of US11 with FcRn HC interferes with FcRn binding to IgG.

FcRn transports IgG across the polarized epithelial cells[40,42]. HCMV can infect Caco-2 cells at the basolateral membrane[64]. We explored the possibility that HCMV-infected Caco-2 epithelial cells have altered IgG transcytosis. We added human IgG Abs to the apical surface of a Caco-2 cell monolayer at 37 °C and then measured IgG transport into the opposite basolateral chamber using cells infected with HCMV in the basolateral surface. After 2 h, IgG transport was significantly ($P < 0.001$) decreased or blocked in HCMV-infected Caco-2 cells (Fig. 8e, lane 2, Fig. 8f) when compared to mock-infected cell monolayers. We also showed that IgG transport was significantly decreased ($P < 0.001$) or blocked in HCMV-infected placental epithelial BeWo cells (Fig. 8g, lane 3; Fig. 8h) as compared with mock-infected BeWo cell monolayers (lane 1).

Infecting Caco-2 monolayers with HCMV for 48 h may result in a leakage. To further show whether US11 expression alone reduces IgG transcytosis, we stably expressed US11 in Caco-2 cells (Supplementary Fig. 9a). Then, IgG (0.5 mg/ml) was added apically into the Caco-2$^{US11}$ cell monolayer and further incubated for 2 h to allow for transcytosis. We then collected the basolateral medium and measured IgG. As seen in Caco-2$^{US11}$ cells, IgG transport was significantly decreased ($P < 0.001$) (Fig. 8i, lane 2; Fig. 8j) as compared with mock-transfected cells (Fig. 8i, lane 4;

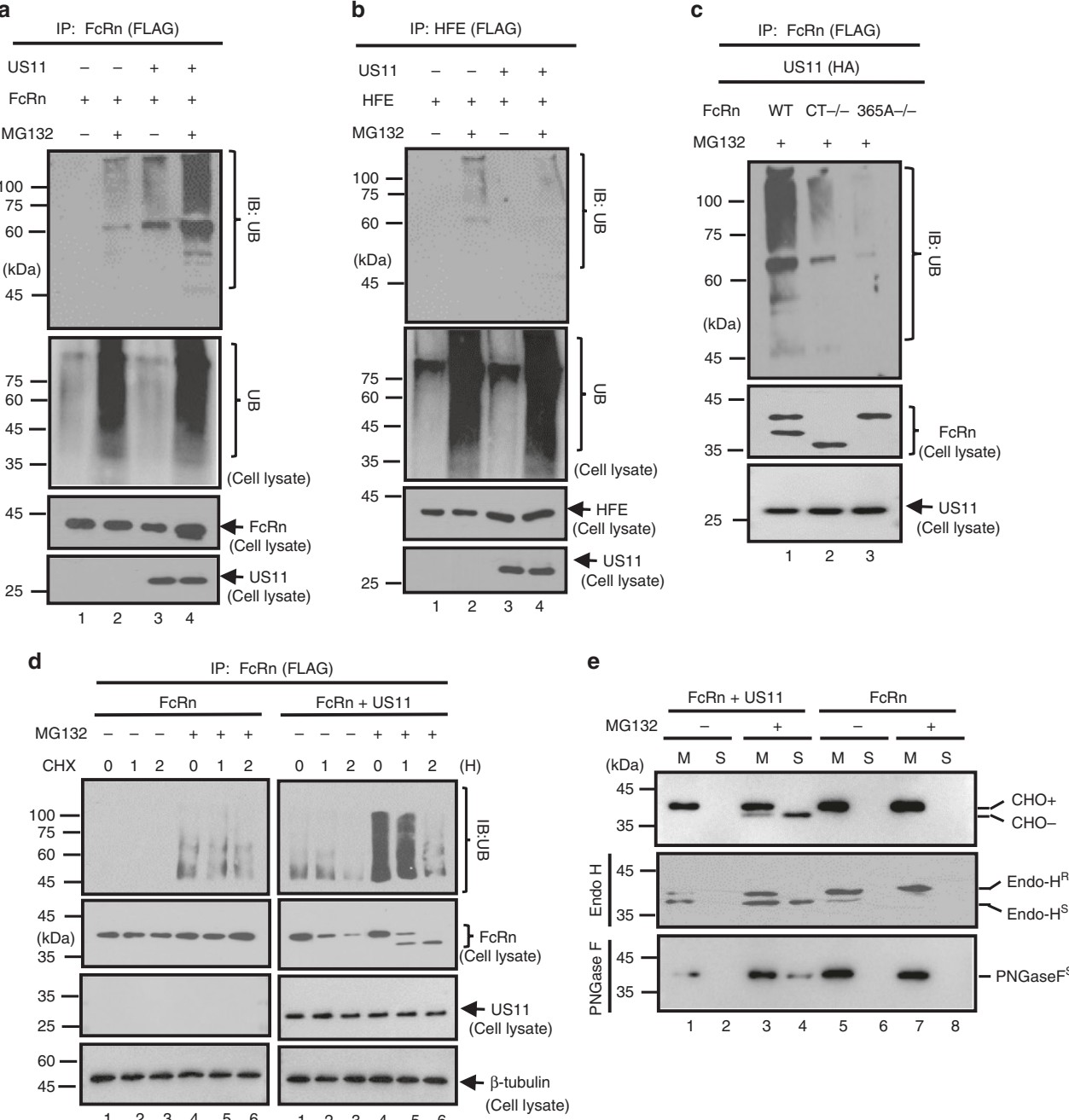

**Fig. 6** US11 induces FcRn dislocation and ubiquitylation. **a–d** FcRn is ubiquitinated in the presence of US11 and MG132. HeLa[FcRn] (**a**), HeLa[HFE] (**b**), HeLa[FcRn] CT−/− (**c**, lane 2), and HeLa[FcRn 365A−/−] (**c**, lane 3) cells were transfected with or without US11 plasmids for 48 h, and cells were treated with proteasome inhibitor MG132 (50 μM) for 2 h, as indicated. HeLa[FcRn] or HeLa[FcRn+US11] cells (**d**) were treated with CHX (100 μg/ml) and chased for the indicated time in the presence of MG132. Cell lysates (0.5 mg) were immunoprecipitated with mAb anti-FLAG for FcRn (**a**, **c**, **d**) or HFE (**b**). All immunoprecipitates or cell lysates (20 μg) as internal controls were subject to western blotting with corresponding Abs as indicated. **e** Fractionation of FcRn HC. HeLa[FcRn], HeLa[US11+FcRn] cells were incubated in the presence or absence of 50 μM MG132 for 4 h. Cells were then homogenized and the homogenates were fractionated by centrifugation (see Methods). FcRn in the membrane pellet (M, lanes 1, 3, 5, 7) and supernatant (S, lanes 2, 4, 6, 8) fraction was digested by mock (top), Endo-H (middle), PNGase F (bottom) enzymes for 2 h at 37 °C, respectively. R: resistant; S: sensitive. After treatment all fractions were subject to western blotting with corresponding Abs as indicated

Fig. 8j). IgG was not transported at 4 °C in HCMV-infected Caco-2 (Fig. 8e, lane 4) or BeWo cells (Fig. 8g, lane 4), and US11-expressing Caco-2 cells (Fig. 8i, lane 1), suggesting that transepithelial flux of IgG Abs by passive diffusion across intercellular junctions or monolayer leaks did not contribute to the amount of detected IgG. Hence, we conclude that US11 inhibits IgG transport across polarized epithelial cells.

**US11 expression facilitates IgG degradation**. FcRn plays a critical role in IgG homeostasis[39,65–67]. If US11 acts by preventing FcRn binding to IgG and trafficking to the endosome, it should consequently accelerate IgG degradation by promoting the trafficking of pinocytosed IgG to the lysosomes for degradation. To test this hypothesis, HCMV-infected endothelial HMEC-1 cells (Supplementary Fig. 9b) were incubated with IgG. We found that

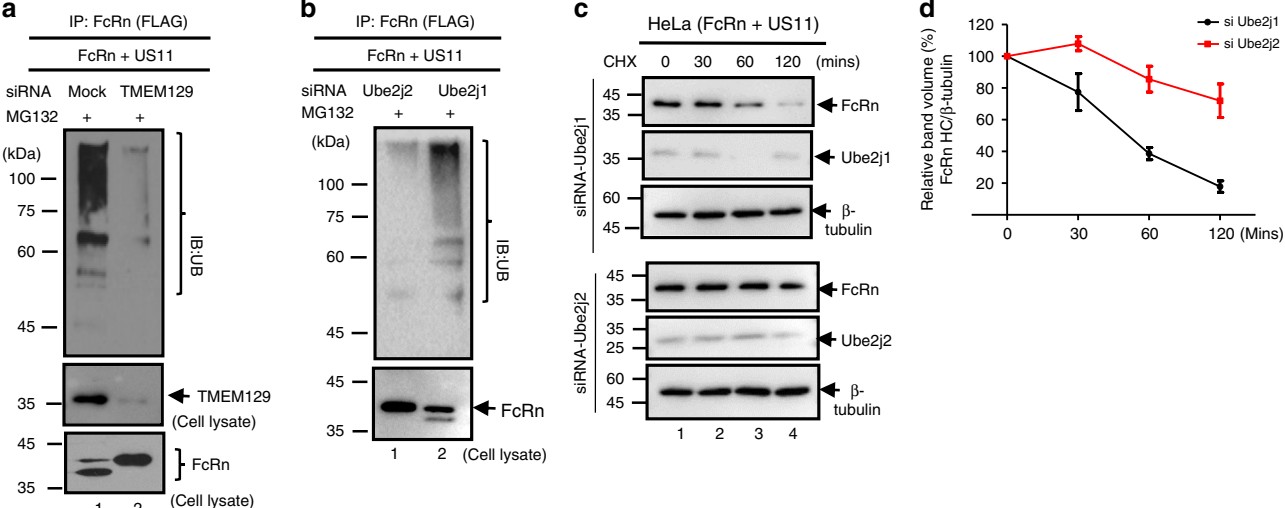

**Fig. 7** TMEM129 and Ube2J2 are required for US11-induced FcRn ubiquitination. **a, b** TMEM129 and Ube2J2 is required for US11-induced FcRn ubiquitination. HeLa$^{FcRn+US11}$ cells were transfected with 20 nM TMEM129, Ube2J1, or Ube2J2 siRNA oligomers for 48 h or empty vector. Cells were subsequently treated with 50 μM MG132 for 4 h and then lysed. All immunoprecipitates or cell lysates (20 μg) as internal controls were subject to western blotting with corresponding Abs as indicated. **c, d** HeLa$^{US11+FcRn}$ cells were transfected with 20 nM Ube2J1 (**c**, top) and Ube2J2 (**c**, bottom) siRNA oligomers for 48 h. Cells were then treated with CHX (100 μg/ml) and chased for the indicated time. The cells were lysed after CHX treatment and cell lysates (20 μg) were subject to western blotting with Abs as indicated. The level of remaining FcRn (**d**) in Ube2j1 siRNA-treated cells (black) or Ube2j2 siRNA-treated cells (red) was quantified as the percentage of the β-tubulin level at different time points. The percentage of time point 0 (min) is assigned a value of 100% and the values from other time points are normalized to this value. Error bars represented mean ± SEM of three independent repeats

after a 48 h incubation, the level of IgG that was recycled back to the supernatant was significantly reduced ($P < 0.01$) in HCMV-infected HMEC-1 cells in comparison with that in mock-infected cells (Fig. 9a, b). To further elucidate this process, we stained HCMV-infected HMEC-1 cells with anti-EEA1 mAb to visualize IgG trafficking to the endosome. We detected IgG in the endosome of mock-infected HMEC-1 cells, but much less in the HCMV-infected cells (Fig. 9c, d). To further investigate the fate of IgG, lysosome-associated membrane glycoprotein-1 (LAMP-1), a lysosomal marker, was used to track IgG trafficking to lysosomal sites. Lysosomal transport of IgG was negligible in mock-infected HMEC-1 cells during the incubation period (Fig. 9e). However, co-localization (Fig. 9e, *yellow*) of LAMP-1 and IgG was more prominent in HCMV-infected HMEC-1 cells, suggesting that HCMV infection promotes lysosomal trafficking of IgG. Pearson's correlation coefficient analysis indicated significant co-localization of IgG with endosomal (Fig. 9d) but much less lysosomal (Fig. 9f) markers in mock-infected HMEC-1 cells. IgG trafficking patterns in HCMV-infected HMEC-1 cells were verified in IgG-treated HeLa$^{FcRn+US11}$ cells (Supplementary Fig. 10). The IgG degradation accelerated by US11 was further verified in IgG recycling experiments, as HCMV infection or US11 expression significantly reduced the IgG recycling in either HMEC-1 cells or HeLa$^{FcRn}$ cells (Supplementary Fig. 11). Taken together, these data strongly suggest that US11 prevents FcRn endosomal trafficking, ultimately resulting in the delivery of IgG to lysosomes for degradation.

## Discussion
Misfolded proteins are translocated across the ER-membrane, destined for cytosolic proteasome degradation in a process known as ERAD[68]. Many pathogens including HCMV, exploit the ERAD system to nullify important components of the host immune system[69,70]. The success of HCMV to infect a large proportion of the world's population is due at least in part to its ability to evade the cellular immune system, and we have found that the virus also

inhibits antibody responses. We initially hypothesized that HCMV infection impeded FcRn trafficking from the ER, leading to its absence in the acidic endosomal compartment and rendering it unable to bind IgG. In this study, we demonstrate that HCMV infection causes FcRn degradation through an ERAD mechanism and thus, removes a critical feature of host immunity.

We first determined that the HCMV protein US11 colocalizes with FcRn in transfected or HCMV-infected cells. This observation led us to measure a specific interaction between US11 and FcRn in human epithelial, endothelial, and macrophage cells. Our results in THP-1 cells were especially encouraging, because cells in the monocyte lineage are known to be a reservoir for latent HCMV infection. We subsequently identified early immature glycosylation patterns on the US11–FcRn complex, suggesting that the complex is retained in the ER and supporting the observation that US11 is an ER resident protein[71]. We then observed that the presence of US11 correlates with an absence of FcRn in early endosomes. We also determined that US11 interacts with nascent FcRn HC but not with FcRn-$\beta_2$m, suggesting that US11 captures newly synthesized FcRn HC before it binds $\beta_2$m. The exact molecular interactions between US11 and FcRn have yet to be fully characterized, but it seems that the US11 luminal domain targets the extracellular domain of FcRn, and we will investigate the contribution of a critical residue to this interaction. We found the evidence that the HCMV genome encodes other proteins that interact with FcRn HC. These additional HCMV proteins may affect FcRn function and make the functional studies more complexed in using US11 mutant virus.

Interestingly, we observed no degradation of FcRn tailless, even in the presence of US11 binding, suggesting that US11 is not sufficient for FcRn degradation and the FcRn cytosolic tail is necessary for trafficking into the degradative ERAD pathway. We discovered that US11 interacts with Derlin-1 that was capable of causing FcRn degradation. We also observed that deletion of the Derlin-1 cytosolic tail also prevented it from tightly binding to FcRn HC. We postulate that the interaction of the FcRn and Derlin-1 cytoplasmic tails may allow both molecules to engage

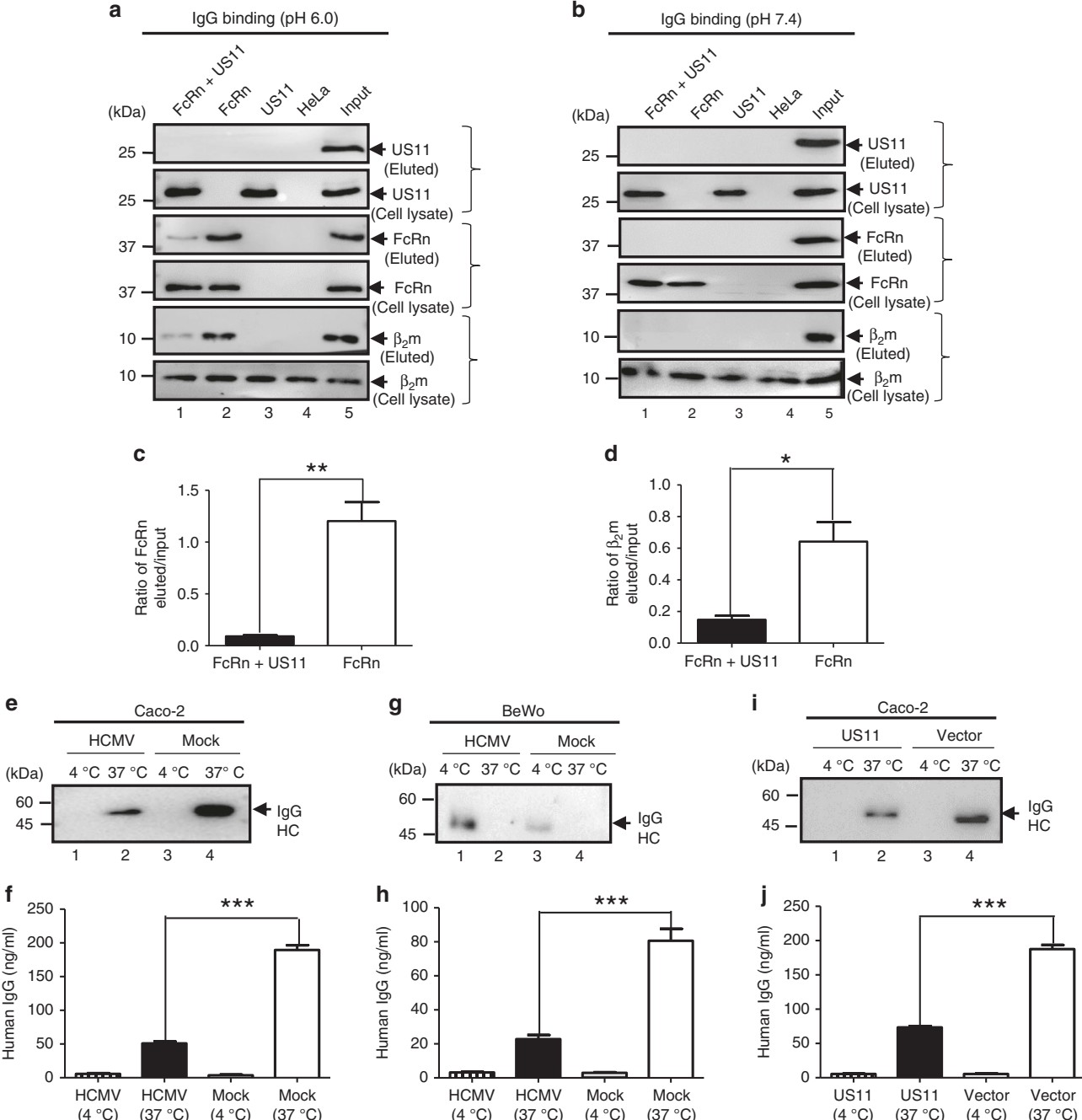

**Fig. 8** HCMV and US11 reduce FcRn-mediated IgG transcytosis in epithelial monolayers. **a–d** The presence of US11 reduces FcRn binding to IgG. HeLa transfectants were lysed in pH 6.0 (**a**) or pH 7.4 (**b**) condition. Approximately 0.5 mg of soluble proteins were incubated with human IgG-Sepharose at 4 °C. Eluted proteins were subjected to western blotting analysis with corresponding Abs as indicated. Cell lysates were blotted as control. The amount of eluted FcRn (**c**) or $\beta_2$m protein from HeLa$^{FcRn}$ and HeLa$^{FcRn+US11}$ cells was compared by the ratio of the band density of eluted protein to that of input protein. The density of protein bands was quantified by the Image Lab 5.2 software. **e–h** Caco-2 cells ($2 \times 10^4$/well) or BeWo cells ($10^5$/well) were grown in 0.4 μm transwell for 8 to 10 days (Caco-2) or for 4 days (BeWo) to allow differentiation. When the transepithelial resistance $r$ reached above 600 (Caco-2) or 400 (BeWo) ohms/cm$^2$, cells were infected at the basolateral surface with HCMV (MOI 5) for 1 h. After washing, cells were incubated for 48 h. Infected or mock-infected cells were loaded at the apical surface with IgG (lanes 1–4) (0.5 mg/ml for Caco-2 or 0.25 mg/ml for BeWo) at 37 °C or 4 °C, respectively. Medium was collected from the basolateral compartment 2 h later and subjected to western blot (**e, g**) or ELISA (**f, h**) analysis. **i, j** Caco-2 cells transfected with either pEF6 or pEF6-US11 plasmid were grown on transwell inserts. The cells were incubated for 1 h at 37 °C or 4 °C, then IgG (0.5 mg/ml) was added to the apical surface and further incubated for 2 h to allow transcytosis. Medium from the basolateral compartment was collected and IgG content was measured by western blot (**i**) or ELISA (**j**). *$P < 0.05$, **$P < 0.01$, and ***$P < 0.001$ by Student's $t$-test. Error bars represented mean ± SEM of three independent repeats

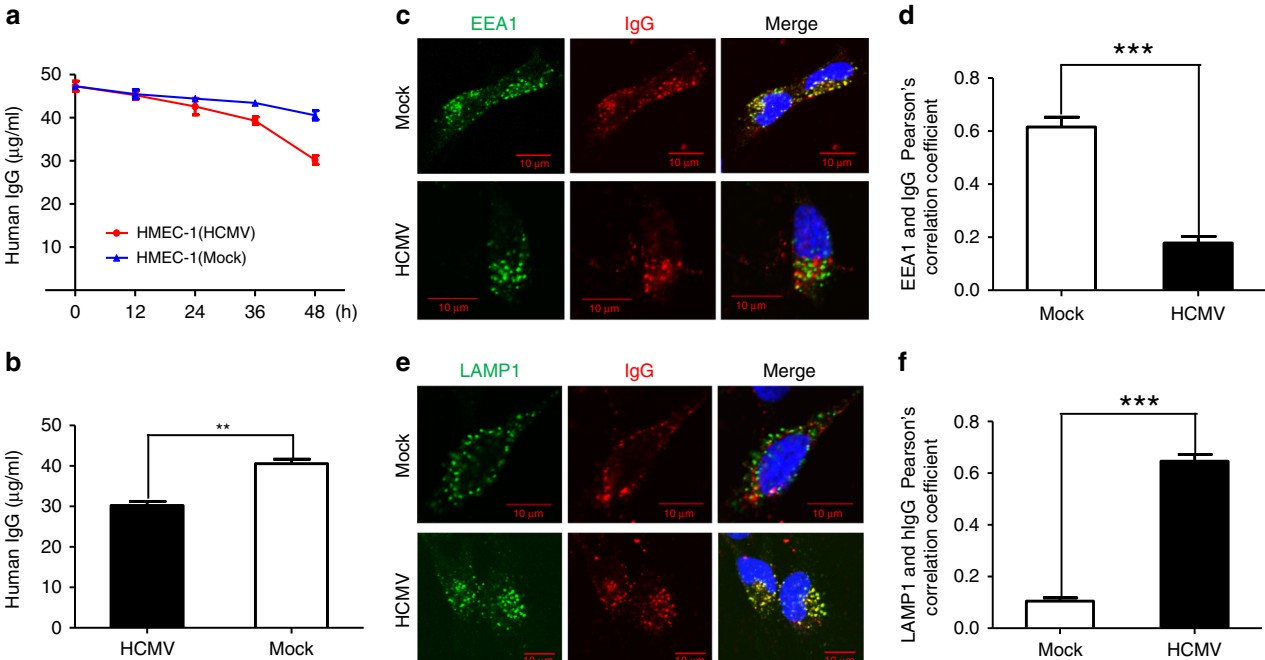

**Fig. 9** HCMV infection increases IgG catabolism in human endothelial cells. **a**, **b** HEMC-1 cells ($5 \times 10^5$/2 ml) were grown in complete medium with 5% FBS with ultra-low IgG. After cells were infected with 5 MOI of HCMV or mock-infected for 48 h, they were incubated with 50 μg/ml human IgG for 48 h at 37 °C. After washing, the cells were incubated at 37 °C. During the incubation, 50 μl of supernatant was sampled at 0, 12, 24, 36, and 48 h and the IgG concentration in each sample was measured by ELISA (**a**). At 48 h, the IgG concentration in the medium from the HCMV-infected and mock-infected cells was analyzed by t-test (**b**). **P < 0.01 by Student's t-test. Error bars represented mean ± SEM of three independent repeats. **c–f** To visualize human IgG trafficking inside infected HEMC-1 ($5 \times 10^4$) cells, cells were infected with 5 MOI of HCMV for 48 h and incubated with 250 μg/ml human IgG for 1 h at 37 °C. After washing, cells were incubated in complete medium without IgG for an additional 1 h, then fixed and stained for co-localization of human IgG with the early endosomal marker EEA1 (**c**) or lysosomal marker LAMP-1 (**e**). Scale bar: 10 μm. For Pearson's correlation coefficiency measurement, ten microscopic fields, each of which contained at least ten cells, were measured for correlation coefficiency rate (**d**, **f**). ***P < 0.001 by Student's t-test. Error bars represented mean ± SEM of three independent repeats

ER dislocation machinery. We observed that FcRn, following its capture by Derlin-1, is quickly dislocated, ubiquitinated and released from the ER into the cytosol. Although Derlin-1 interacts with a range of E3 ligases, its durable interaction with TMEM129 made TMEM129 a strong candidate for the final ubiquitination of FcRn. Structural analysis of TMEM129 identified it as a possible member of the RING family of E3 ligases[60,61]. We demonstrated that the knockdown of TMEM129 and its cognate E2 UbE2J2 abolishes FcRn ubiquitination and prevents its degradation, even in the presence of US11. We suggest that the major role of US11 is to facilitate FcRn binding to Derlin-1, leading to TMEM129-mediated ubiquitination of FcRn and its subsequent proteasomal degradation in the cytosol. Thus, US11 harnesses a Derlin-1/TMEM129-dependent pathway that is responsible for FcRn degradation. HCMV downregulated FcRn expression in human foreskin fibroblast (HFF), which was detected by mass spectrometry[23]. However, we failed to detect FcRn expression by western blot analysis in either uninfected or HCMV-infected HFF (Supplementary Fig. 12). This discrepancy may be caused by the protein detection method or the low level of FcRn expression in the HFF cells. More experiments are needed to verify this result[23].

We characterized the effects of US11 on FcRn-mediated IgG transport and protection. As HCMV infection cycles through periods of latency and reactivation, the long-lasting humoral immune responses are generated through the production of IgG. FcRn normally supports passive immunity in the neonate by facilitating transfer of maternal IgG across the placenta. Post delivery, FcRn continues to support an effective immune response against infection by transporting IgG across polarized epithelium

lining mucosal surfaces[40–42]. FcRn therefore contributes to a lifelong IgG-mediated immunity. For HCMV to evade immunity and be shed despite a potent immune response, we suggest that it must have developed a mechanism to inhibit FcRn. We found that expression of US11 in cells reduced the ability of FcRn to transport IgG across epithelial surfaces and resulted in reduced half-life of the IgG. This observation could help explain the association of HCMV with severe congenital infection[72]. We propose that US11 may help HCMV avoid contact with IgG by accelerating the catabolism of FcRn. This property of HCMV would be especially important after virus reactivation in order to maintain long-term infection and shedding in its host.

FcRn is normally glycosylated. These glycation moieties may inhibit entry into the proteasome. We postulate that N-glycanase likely removes these glycans following FcRn dislocation from the ER. During our studies, we detected a deglycosylated form of FcRn only in the cytosol (Fig. 6e). As the deglycosylated FcRn intermediates are not observed in the absence of proteasomal inhibitors, this suggests that the dislocation from the ER and proteasomal degradation may be tightly coupled. This assumption is supported by evidence that the C-terminal region of Derlin-1 interacts with the cytosolic proteasomal protein AAA ATPase p97[73,74]. Therefore, we suspect that p97 may interact with Derlin-1-bound FcRn HC to provide the activation energy necessary for FcRn extraction from the ER membrane into the cytosol for proteasomal degradation. Our results imply that the cytosolic region of FcRn is involved in ERAD substrate binding and this interaction is critical for the Derlin-1-mediated dislocation of FcRn to the cytosol during US11-induced FcRn degradation.

US11 has previously been shown to degrade MHC class I molecules through a Derlin-1/TMEM129-mediated ERAD system, thereby hindering the recognition of infected cells by CD8+ cytotoxic T cells[11,75]. We have verified this finding in our study (Supplementary Figs. 13 and 14). However, our discovery of US11-mediated FcRn degradation is unexpected because FcRn has limited homology with MHC-I. Previous studies reported the HCMV protein US2[76] can also induce rapid degradation of newly synthesized MHC-I. In contrast, HCMV US2 fails to induce FcRn degradation. We suspect that HCMV is likely to possess multiple immune-modulating proteins to degrade MHC-I molecules due to their highly polymorphic nature but FcRn is relatively non-polymorphic. Most surprisingly, we have found that this US11-mediated ERAD degradation mechanism evades both cellular and humoral immunity, thus providing HCMV an efficient option to evade the human immune responses in a broad sense.

Several questions remain for future studies. First, the precise sites of ubiquitination in US11-induced FcRn degradation remain undefined. Human FcRn contains a single lysine residue in its cytoplasmic tail[55]—whether this lysine residue is critical for FcRn ubiquitination merits further investigation. Ubiquitination is known to occur at nonlysine residues, including serine, threonine, and cysteine[77,78]. Both serine and threonine residues appear in the FcRn CT[55], leading to the possibility of multiple ubiquitin acceptor sites. In addition, TMEM129 may be able to ubiquitinate a combination of lysine and nonlysine residues. US11 is observed to degrade MHC-I molecules that do not contain lysine residues[79], further supporting our notion that TMEM129-dependent ubiquitination of FcRn HC may occur on a combination of lysine and nonlysine residues. Second, we propose that US11 inhibits intracellular trafficking of FcRn, leading to decreased IgG transcytosis and increased IgG catabolism. These mechanisms cannot be currently tested in vivo as HCMV is highly species-specific and no animal model is available. Cytomegalovirus strains infect mice and guinea pigs and it will be interesting to know whether these strains also cause FcRn degradation. Third, FcRn has an important role in the development of autoimmune disease[80] because it prolongs the half-life of autoreactive IgG. Thus, HCMV US11 might serve as a treatment for patients with autoimmune disease by blocking FcRn function and facilitating the destruction of autoreactive IgG.

In conclusion, we define a mechanism by which US11 suppresses humoral immunity through the inhibition of FcRn. We therefore introduce a mechanism for the role of US11 as a humoral immune suppressor. We propose a model that details how HCMV US11 dislocates FcRn to the cytosol for subsequent proteasomal degradation (Fig. 10). Our characterization of HCMV US11-induced FcRn ubiquitination as a target by the ERAD system therefore not only uncovers a function for the ERAD pathway, but may better help in understanding HCMV pathogenesis, treating viral diseases, and designing effective vaccines. Owing to the global prevalence of HCMV infection and the important roles for FcRn in IgG transport and catabolism, this study may impact work in multiple fields, including infectious disease, rheumatology, and oncology.

## Methods

**Cell lines, enzymes, and viruses**. Human intestinal epithelial Caco-2, human placental trophoblast BeWo, human lung fibroblast MRC-5, cervical HeLa cell lines were obtained from the American Type Culture Collection (ATCC, Manassas, VA) and grown in complete DMEM. The human endothelial cell line HMEC-1, a dermal-derived microvasculature cell line, was provided by the Centers for Disease Control (Atlanta, GA). Human monocytic THP-1 cells (ATCC) and HMEC-1 cells were grown in RPMI 1640 (Invitrogen, Carlsbad, CA) complete medium. Complete media were supplemented with 10 mM HEPES, 10% FCS (Sigma-Aldrich, St. Louis, MO), 1% L-glutamine, nonessential amino acids, and 1% penicillin-streptomycin. To differentiate THP-1 cells into macrophages, cells were treated with 50 ng/ml

phorbol-12-myristate-13-acetate (PMA) for 48 h. Primary human umbilical vein endothelial cells (HUVEC, ATCC PCS-100-013™) were purchased from ATCC and grown in vascular cell basal medium (PCS-100-030™) supplemented with endothelial cell growth kit-BBE (PCS-100-040™). Cells were grown in 5% CO₂ at 37 °C.

The HCMV AD169 strain was purchased from ATCC. An HCMV clinical strain, designated for CMV Bethesda BAL, was isolated from bronchoalveolar lavage fluid in a patient who signed consent on an Institution Review Board approved protocol (01-I-0161) at the National Institute of Allergy and Infectious Diseases, National of Institutes of Health, Bethesda, MD. The virus was passaged <5 times in MRC-5 cells before use in these experiments. Ten to fourteen day post infection, cell-free virus was harvested by sonicating cell pellets, and the cellular debris was removed by centrifugation at 6000 relative centrifugal force (rcf) for 20 min. Virus was concentrated by centrifugation at 20,000 rcf for 2 h through a 20% sucrose cushion. Virus titer was determined by quantifying TCID₅₀ in MRC-5 (ATCC) cells using the Reed-Muench method. HCMVΔUS1-US12[81] was a kind gift from Dr. Fenyong Liu, University of California, Berkeley and Dr. Hua Zhu, New Jersey Medical School, Rutgers University.

**Antibodies and fluorescent dyes**. The following antibodies were used for detection of endogenous and overexpressed proteins by western blot analysis: rabbit anti-Flag epitope (DYKDDDDK) (Sigma-Aldrich, cat# F7425; 1:5000), rat anti-FLAG epitope (Biolegend, Cat# 637301; 1:5000), rat anti-HA epitope (YPYDVPDYA) (Roche, cat# 11867423001; 1:5000), mouse anti-Myc epitope (EQKLISEEDL) (Cell Signaling, Cat# 2276S; 1:5000), rabbit anti-Myc epitope (Cell Signaling, Cat# 2278T; 1:2500), mouse anti-human FcRn (Santa Cruz Biotechnology, Cat# Sc-271745; 1:500), rabbit anti-human FcRn (affinity purified, 1:500), mouse anti-US11 (Affinity-purified, 1:500), mouse anti-human β₂m (hybridoma BBM1; 1:500), rabbit anti-human β₂m (Abcam, Cat# ab75853; 1:1000), rabbit anti-human β-tubulin (Sigma-Aldrich, Cat# T2200-200UL; 1:2500), mouse anti-HCMV PP65 (Abcam, Cat# ab6503; 1:2500), mouse anti-ubiquitin (Santa Cruz Biotechnology, Cat# Sc-8017; 1:250), rabbit anti-human TMEM129 (Sigma-Aldrich, Cat# SAB1302253; 1:500), mouse anti-human MHC class I (Enzo Life Sciences, Cat# ALX-805-711-C100; 1:500), rabbit anti-human CD71 (TfR1) (Santa Cruz Biotechnology, Cat# sc-9099; 1:500), mouse anti-human Ube2j1 (Santa Cruz Biotechnology, Cat# sc-3777002; 1:500), rabbit anti-human Ube2j2 (Novus biological, Cat# NBP1-59760 1:500). For detection by western blot, primary antibody incubation was followed by incubation with appropriate secondary HRP-conjugated goat anti-mouse/rabbit/rat/human (Southern Biotech, Cat# 1010-05/ 4030-05/3030-05/2081-05, respectively) in a 1:5000 dilution. The following antibodies against human antigens were used for immunoprecipitation in a 1:100 dilution: rabbit anti-FLAG epitope (Sigma-Aldrich, cat# F7425), rat anti-FLAG epitope (Biolegend, Cat# 637301), mouse anti-HA epitope (hybridoma 12CA5, affinity purified), mouse anti-Myc epitope (Cell Signaling, Cat# 2276S), rabbit anti-human FcRn (affinity purified), mouse anti-human β₂m (hybridoma BBM1), rabbit anti-human TMEM129 (Sigma-Aldrich, Cat# SAB1302253). The following antibodies were used for Flow cytometry: rat anti-FLAG epitope (Biolegend, Cat# 637301; 1:500), mouse anti-human FcRn (Santa Cruz Biotechnology, Cat# Sc-271745; 1:500). For detection by confocal microscopy, primary antibody incubation 1 h on ice was followed by incubation with appropriate secondary goat anti-mouse Alexa 488 (Life Technologies, Cat# A11001), goat anti-rat Alexa 488 (Life Technologies, Cat# A11006) in a 1:1000 dilution. The following antibodies were used for confocal microscopy: rabbit anti-Flag epitope (Sigma-Aldrich, Cat# F7425; 1:500), rat anti-HA epitope (Roche, Cat# 11867423001; 1:500), mouse anti-human EEA1 (BD Biosciences, Cat# 610457; 1:250), mouse anti-human LAMP-1 (BD Biosciences, Cat# 611042; 1:250), rabbit anti-pp65 (biorbyt, Cat# orb10511; 1:250), mouse anti-US11 (Affinity-purified, 1:200). For detection by confocal microscopy, primary antibody incubation 1 h at room temperature was followed by incubation with appropriate secondary goat anti-mouse Alexa 488/555 (Life Technologies, Cat# A11001/A21424, respectively), goat anti-rabbit Alexa 555 (Life Technologies, Cat# A21430), goat anti-rabbit Alexa 488 (Abcam, Cat# ab150077), goat anti-rat Alexa 488 (Life Technologies, Cat# A11006), goat anti-human Alexa 555 (Life Technologies, Cat# A21433) in a 1:1000 dilution. Goat anti-Human IgG Fc coating antibody (Bethyl, Cat# A80-104P; 1:100) and goat anti-Human IgG Fc HRP-conjugated antibody (Bethyl, Cat# A80-104P; 1:10000) were used for ELISA.

**Viral infection of human cells**. HeLa, THP-1, HMEC-1, BeWo, primary HUVEC, and Caco-2 cell monolayers were infected with the HCMV clinical strain through the addition of virus at a high MOI of 5 or 10. Prior to infection, cells were washed once with Dulbecco's phosphate-buffered saline (PBS). After the addition of virus, infection proceeded for 2 h at 37 °C in an atmosphere of 5% CO₂. After washing three times with PBS, infected cells were cultured with fresh complete medium at 37 °C in an atmosphere of 5% CO₂.

**Construction of expression vectors and mutagenesis of US11**. All primers used for cloning or mutagenizing the genes in this study are summarized in Supplementary Table 1. The construction of human β₂m and FcRn expression plasmids, pcDNAβ₂m, and pcDNA-FLAGFcRn was previously described[51]. HLA-A₂ cDNA was amplified from HeLa total RNA by RT-PCR and was cloned into pcDNA-Flag via Hind III and Xba I restriction site cloning. Homeostatic iron regulator (HFE)

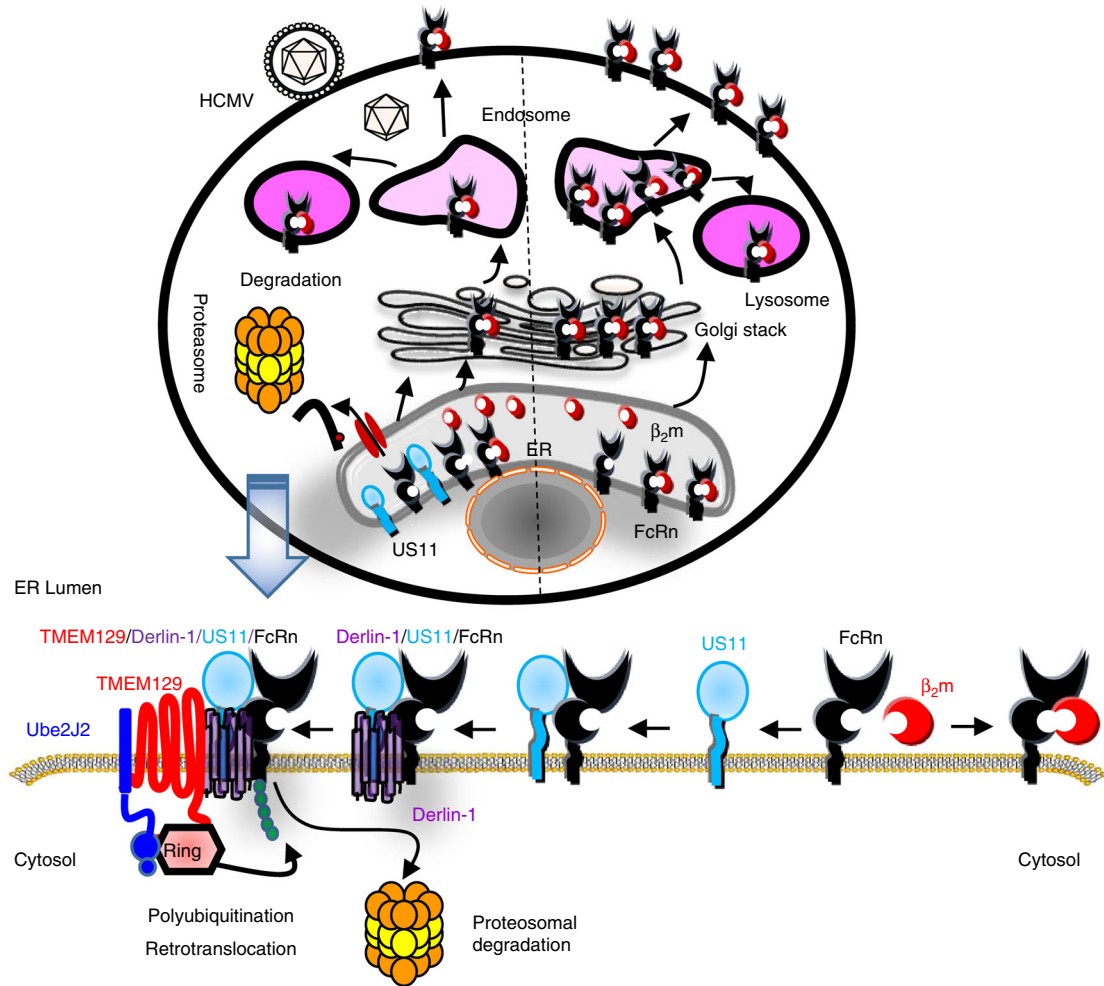

**Fig. 10** Proposed model for US11 interaction with FcRn. In uninfected cells (right), FcRn traffics to the endosome and reaches the cell surface through the secretory pathway and recycles between the plasma membrane and endosomes via endocytosis. In HCMV-infected cells or in cells expressing US11 (left), a portion of the β2m-free FcRn HC molecules is associated with US11 in the ER. US11-bound FcRn is rapidly ubiquitinated by TMEM129 E3 ligase and subsequently dislocated to the cytosol for proteasomal degradation. TMEM129 is recruited to US11 via Derlin-1. The portion of FcRn engaged by US11 is targeted for proteasome degradation by ER 'dislocation''. The Figure was created by Dr. Xiaoping Zhu with assistance of the PowerPoint Drawing Diagrams (http://www.motifolio.com)

encoding the human hemochromatosis protein was amplified from pCMV-Sport-HFE and cloned into pcDNA-Flag using *Hind* III and *Xba* I restriction site cloning. An FcRn mutant without the cytoplasmic tail, FcRn CT−/−, or FcRn mutant deleted for amino acid 365 in its C-terminus, FcRn 365A−/−, were amplified from pcDNA-FLAGFcRn and subsequently cloned into pcDNA-Flag using *Not* I and *Xba* I digestions. To construct pSectag2-Derlin-1, Derlin-1 was amplified from HeLa cDNA and its C-terminus was fused to a Myc epitope. The DNA fragment was digested with *Xba* I and *Xho* I (underlined) and ligated into the plasmid pSectag2, which was pre-digested with *Nhe* I (isocaudomer of *Xba* I) and *Xho* I enzymes. A Derlin-1 mutant deleted for amino acids 1–66 in its N-terminus (NT−/−) or deleted for amino acids 526–756 in its C-terminus (CT−/−) was amplified from pSectag2-Derlin-1 and subsequently cloned into pSecTag Hygro A plasmid using *Xba* I and *Xho* I double digestions. The pTFR1-GFP plasmid was a gift from Dr. Gary Banker (Oregon Health and Science University, Portland, OR).

The purified HCMV AD169 DNA was only used as a template for synthesis of HCMV genes, US11 or US2. In brief, the pEF6-US11 and pEF6-US2 constructs were constructed by fusing an HA epitope to the N-terminus of either HCMV US11 or the C-terminus US2 gene by the PCR primer pairs listed in Supplementary Table 1. The N-terminal HA tag was inserted between the US11 signal peptide and the US11 ORF. All DNA fragments were digested with *BamH* I and *Xba* I (underlined) and ligated into the plasmid pEF6 to generate the plasmid pEF6-US11 or pEF6-US2. A US11 mutant was generated by mutation of a polar amino acid, glutamine (Q) 192, within the US11 transmembrane domain to a hydrophobic leucine (L) residue using a site-directed mutagenesis kit (Takara, Mountain View, CA). The US11 DNA in the pEF6 expression vector was used as a template. The oligonucleotide was used for the change of a glutamine (Q) 192 to a leucine (L) residue, base substitutions are underlined. The resultant plasmids were designed for pcDNAUS11Q192L. To construct a pGex4T1-US11, a PCR primer pair was used to

amplify a truncated 438 bp DNA fragment encoding the extracellular domain of US11 gene. In the above cloning, the primer introduced a *BamH* I or *Not* I site (underlined) to facilitate subcloning of the DNA fragment into the pGEX4T-1 (Amersham Pharmacia Biotech, Piscataway, NJ) expression vector. All constructs were sequenced to verify the fidelity of amplification, cloning, and mutations. All oligonucleotides used in this study are summarized (Supplementary Table 1).

**Production of FcRn-specific and US11-specific antibody.** Production of affinity-purified FcRn-specific Abs was previously described[53]. Production of affinity-purified glutathione S-transferase (GST) fusion proteins was done as previously described[53]. In brief, recombinant GST-US11 proteins were produced in BL21 cells (Invitrogen) following treatment with 0.2 mM IPTG (isopropyl-β-D-thiogalacto-pyranoside) for 16 h. To produce anti-US11 antibodies, we immunized a mouse with the purified GST-US11 fusion protein and Freund's adjuvant. Anti-US11 antibodies were then affinity purified from the immunized mouse sera. The animal experiment was approved by the University of Maryland Institutional Animal Care and Use Committee.

**Semiquantitative RT-PCR analysis.** Semiquantitative RT-PCR was performed according to the manufacturer's instructions. In brief, Caco-2 cells ($5 \times 10^5$) were infected with HCMV (MOI 5) or mock-infected for 48 h, and then cells were treated with CHX (100 μg/ml) from 30 to 240 min. Total RNA was isolated using TRIzol (Invitrogen). First-strand cDNAs were obtained from total RNA (100 ng) using a SuperScript™ III Reverse Transcriptase kit (ThermoScientific). Then the cDNAs were used as the template for PCR amplification by human FcRn (5′-GTACCTGAGCTACAATAGCCTG-3′, 5′-CACGGAAAAGCCAGGGCTGCTG-3′ or GAPDH (5′-TGGCGTCTTCACCACCATGGAG-3′, 5′-

AGTTGTCATGGATGACCTTGGCC-3′) specific primers. Twenty-nine or 34 cycles of PCR amplification were performed in a 20-μl volume. Each cycle consisted of denaturation at 94 °C for 30 s, annealing at 58 °C for 30 s, and extension at 72 °C for 30 s. An additional 10 min was applied for the final extension. PCR products were resolved on 1.5% agarose gels and visualized by staining with ethidium bromide. Integrated density values for the FcRn bands were normalized to the GAPDH values to yield a semiquantitative assessment by densitometric intensity analyses.

**Transfection and protein expression**. The stable cell line, HeLa$^{FcRn}$, has been described previously[51]. HeLa, HeLa$^{FcRn}$, and Caco-2 cells were transfected with either empty vector or the recombinant plasmid along with PolyJet transfection reagent (SignaGen Laboratories, Rockville, MD). Single transfectants were selected with G418 (1 mg/ml). Double transfectants were selected with G418 plus either Blasticidin (5 μg/ml) or Hygromycin B (200 μg/ml). Positive clones were tested for protein expression with western blot using anti-FLAG, anti-HA, and anti-Myc antibodies. Successful transfectants were maintained in complete DMEM medium containing G418 (400 μg/ml) ± Blasticidin (3 μg/ml). For transient transfections, cells were transfected with 2 μg of plasmids. The level of protein expression was examined 48 h post transfection by western blot. All transfected cells used in this study are summarized in Supplementary Table 2.

**Silencing of gene expression by siRNA**. Pre-designed siRNA products were synthesized from Genewiz (South Plainfield, NJ), including HCMV US11, human TREM129, and Ube2j1 or Ube2j2 (Supplementary Table 3). Transfection of siRNA oligonucleotides corresponding to US11, TREM129, or Ube2j genes was carried out using Lipofectamine 2000 transfection agent (Invitrogen) at a final concentration of 20 nM mixed siRNA oligomers per well. Mock control was transfected without adding the mixed siRNA oligomers. The US11 and TMEM129 genes were targeted with two non-overlapping siRNAs to enhance effectiveness. For US11 knockdown in primary HUVEC cells by siRNA, cells were transfected with 20 nM siRNA oligomers per well using Nucleofection kit V (Lonza) 24 h before HCMV infection. Knockdown efficiency was confirmed by western blot.

**IgG binding assay**. A human IgG binding assay was performed as previously described[51]. Cells were lysed in PBS (pH 6.0 or 7.4) with 0.5% CHAPS (Sigma-Aldrich) and protease inhibitor cocktail (Calbiochem) mixture on ice for 1–2 h. The soluble proteins (0.5–1 mg) were incubated with human IgG-Sepharose (Rockland Immunochemicals, Pottstown, PA) at 4 °C overnight. Unbound proteins were washed off with PBS (pH 6.0 or 7.4) containing 0.5% CHAPS. Adsorbed proteins were boiled with Laemmli Sample buffer at 95 °C for 5 min. The soluble fractions were subjected to western blot analysis as described below.

**Immunoprecipitation, gel electrophoresis, western blotting**. Cell lines, transfectants, or HCMV- or mock-infected cells ($5 \times 10^6$) were lysed in PBS with 0.5% CHAPS and protease inhibitor cocktail III (Roche, Branchburg, NJ). The cell lysates were centrifuged at 6000 rpm at 4 °C to remove cell debris. Protein concentrations were determined by the Bradford method (Bio-Rad Laboratories). Immunoprecipitations were performed[54]. In brief, the cell lysates (400 μl) were incubated with 100 μl of a protein G agarose slurry plus 5 μg of primary Abs specific for each protein at room temperature for 2 h. The protein G beads were washed by 0.5% CHAPS buffer five times.

The cell lysates or the protein G beads were boiled with 2x Laemmli sample buffer at 95 °C and resolved on a 12% sodium dodecyl sulfate polyacrylamide gel electrophoresis (SDS-PAGE) gel under reducing conditions. Proteins were transferred onto a nitrocellulose membrane (Schleicher & Schuell, Keene, NH) by semi-dry transfer (Bio-Rad Laboratories, Hercules, CA). All blocking, incubation, and washing were performed in 5% non-fat milk and 0.05% Tween 20 in PBS. The membranes were blocked, probed separately with a specific primary Ab, washed, and then probed with an HRP-conjugated secondary Ab for 2 h. Proteins were visualized using immobilon western chemiluminescent HRP substrate (Millipore, Billerica, MA). Chemiluminescence signal acquisition and densitometry analysis were conducted using the Image Lab, version 5.2 in a Chemi-Doc XRS imaging system (Bio-Rad Laboratories, Hercules, CA).

**Analysis of N-linked glycosylation**. N-linked glycosylation was analyzed[54]. In brief, native FcRn in cell lysates or the proteins immunoprecipitated by HA murine antibodies were digested with endo-β-N-acetylglucosaminidase H (Endo H; New England Biolabs) in digestion buffer (100 mM sodium acetate, pH 5, 150 mM NaCl, 1% Triton X-100, 0.2% SDS, 0.5 mM PMSF) or with peptide: N-glycosidase F (PNGase F; New England Biolabs) in 50 mM sodium phosphate, pH 7.5, with 1% NP-40. A mock digestion without enzymes was performed as a control. All digestions were performed for 2 h at 37 °C. Proteins were analyzed on a 12% SDS-PAGE gel under reducing conditions and immunoblotted as previously described.

**Cell fractionation**. Cell fractionation was done as the previously described[82]. After HeLa$^{FcRn}$ and HeLa$^{FcRn+us11}$ cells were incubated with or without MG132 (50 μM)

at 37 °C for 4 h, cells were then pelleted and lysed by three freeze–thaw cycles. Membrane fractions were pelleted from supernatants by ultracentrifugation at 100,000 × g (Beckman XL80, 28,700 rpm) for 2 h. Soluble (cytosolic) fractions were collected and diluted in 1% Triton X-100. Pellet (membrane) fractions were washed by PBS and resuspend in 1% Triton X-100 for further analysis.

**Confocal immunofluorescence**. Immunofluorescence was performed as previously described[53]. Briefly, cells were cultured on coverslips for 24 h at 37 °C. Subsequent procedures were done at room temperature. The cells were rinsed in PBS, fixed in cold 4% paraformaldehyde (Sigma-Aldrich) in PBS for 20 min, and quenched with glycine for 10 min. After two washes with PBS, the coverslips were permeabilized in solution (PBS containing 0.2% Triton X-100) for 5 min and then blocked with blocking buffer containing 3% normal goat serum (NGS) for 30 min. Antibodies diluted in blocking buffer were added onto the coverslips and incubated for 1 h. Cells were then incubated with Alexa Fluor 488 or 555-conjugated goat secondary antibodies in blocking buffer. Cell nuclei were stained with DAPI (4′, 6-Diamidino-2-Phenylindole, Dihydrochloride) for 15 min. After each step, cells were washed three times with 0.1% Tween 20 in PBS. Coverslips were mounted on slides using the ProLong antifade kit (Molecular Probes) and examined using a Zeiss LSM 510 confocal fluorescence microscope. Images were processed using LSM Image Examiner software (Zeiss, Thornwood, NY). Quantitative co-localization measurements were performed using Zeiss LSM 510 Examiner Software. Pearson's correlation coefficient was calculated for describing the co-localization correlation of the intensity distributions between two channels, as previously described[83]. In each quantitative experiment with the transfected HeLa or infected HMEC-1 cells, 100 representative cells were analyzed. A value of $P < 0.05$ was considered significant.

**Quantitative cycloheximide chase assay**. Cells were treated with cycloheximide (CHX) (100 μg/ml) (Calbiochem, San Diego, CA) for different time periods, lysed, and protein levels were measured by Bradford assay. Each cell lysate (20 μg) was analyzed by western blotting with corresponding antibodies. The levels of remaining FcRn, HLA-A2, and β2m at different time points were calculated as the percentage of β-tubulin (an internal control). The percentage of time point 0 (min) was assigned a value of 100% and the values from other time points were normalized to this value. The expression levels of proteins were quantified by the band density (relative band volume) measured by software Image Lab 5.2. The degradation experiments were repeated in triplicate.

**Flow cytometry**. Surface and intracellular expressions of FcRn were examined in either fixed or permeabilized HeLa transfected, THP-1 cells, or HMEC-1 cells by flow cytometry. Cells were washed with PBS, and if necessary, detached by 10% EDTA. For cell surface staining, after blocking by 2% FBS, cells were incubated with rat anti-FLAG Ab for 1 h on ice to minimize internalization. For intracellular staining, cells, were first treated with cycloheximide (100 μg/ml) or left untreated as a control for 4 h. Subsequently, they were incubated with Fixing/Permealizing Buffer (BD CytoFix/CytoPer Kit) for 20 min. THP-1 cells were also treated with 30 μg/ml human Fc block (BD) for 10 min at room temperature. After washing and blocking with 2% FBS, cells were incubated with anti-FLAG Ab or anti-FcRn Ab for 1 h on ice. For all staining, cells were incubated with isotype-matched control rat Abs to determine the background fluorescence. After washing with PBS, the cells were incubated with Alexa Fluor 488-conjugated secondary Abs for 1 h on ice. Cells were fixed with 2% paraformaldehyde overnight and analyzed using a FACSAria (Becton Dickinson, Franklin Lakes, NJ) and FlowJo software (Tree Star).

**Detection of protein ubiquitination in cultured cells**. Cultured cells were transfected with plasmids expressing US11 along with a FLAG-tagged version of FcRn, HFE, and HLA-A2. Forty-eight hours later, cells were treated with 50 μM MG132 (Calbiochem) for 2 h and subsequently lysed in PBS with 0.5% CHAPS and protease inhibitor cocktail. The proteins (500–1000 μg) were incubated with anti-FLAG murine Ab and protein G beads overnight at 4 °C. After the immunoprecipitates were washed three times with PBS containing 0.05% Tween 20 (PBST), they were heated with Laemmli sample buffer at 95 °C and the eluted products were further analyzed by SDS-PAGE and western blot analysis to detect ubiquitin and the targeted proteins with respective antibodies.

**Enzyme-linked immunosorbent assay**. Human IgG was quantified using an enzyme-linked immunosorbent assay (ELISA) kit (Bethyl Laboratories, Montgomery, TX). ELISA plates (Nalge Nunc, Rochester, NY) were coated with goat anti-human IgG-Fc Ab (10 μg/ml) overnight at 4 °C. Plates were washed three times with PBST and then blocked with 2% FBS in PBS for 1 h at room temperature. Plates were washed with PBST three times and incubated with either an IgG standard or the transcytosis samples diluted in 2% FBS for 2 h at 25 °C. Plates were washed for five times with PBST and incubated with HRP-conjugated goat anti-human IgG-Fc Ab (0.1 μg/ml) for 1 h. After plates were washed with PBST seven times, tetramethylbenzidine and hydrogen peroxide were added to initiate the colorimetric reaction; 100 μl of 1 M sulfuric acid was added to stop the reaction.

The colorimetric reaction was read at 450 nm using a Victor III microplate reader (Perkin Elmer, Bridgeville, PA).

**In vitro human IgG transcytosis**. IgG transcytosis was performed as previously described[40–42]. BeWo cells, Caco-2 cells or Caco-2 cells transfected with either pEF6 alone or pEF6-US11 were grown on 0.4 µm Transwell filter inserts (Corning Costar, Corning, NY) to form a monolayer that exhibited a transepithelial electrical resistance (TEER) of 600 ohms/cm$^2$ for Caco-2 cells and 400 ohms/cm$^2$ for BeWo cells, measured using planar electrodes (World Precision Instruments, Sarasota, FL). Prior to infection, Caco-2 or BeWo cell monolayers were washed twice with PBS and then were mock-infected or infected with the HCMV clinical strain at an MOI of 10 for 2 h. After washing, cells were incubated for 48 h at 37 °C in an atmosphere of 5% $CO_2$. TEER was assessed immediately after adding fresh complete medium to verify that monolayers had remained intact during the infection procedure. Human IgG was added to the apical surface of the cells at a final concentration of 0.5 mg/ml (Caco-2 cells) or 0.25 mg/ml (BeWo cells) and monolayers were incubated for 2 h at either 4 °C or 37 °C. For detecting human IgG, an aliquot of the basolateral medium was concentrated using a 0.5 ml Amicon Ultra 10 K centrifugal filter (Millipore, Billerica MA). ELISA was used to quantify human IgG according to the manufacturer's instructions (Bethyl Laboratories, Montgomery, TX). Transported IgG proteins were analyzed by western blot-ECL or ELISA.

**In vitro human IgG protection**. Human IgG protection assay was performed in either HEMC-1 cells ($2.5 \times 10^5$/ml) or HeLa$^{FcRn+US11}$, HeLa$^{FcRn}$, and HeLa cells ($10^6$) that were cultured in complete medium containing 5% FBS with ultra-low IgG. After cells were infected with HCMV at an MOI of 5 or control for 48 h, they were incubated with 50 µg/ml of IgG for additional 48 h. The supernatant (50 µl) was subsequently sampled at 0, 12, 24, 36, and 48 h and stored at 4 °C for ELISA. To visualize IgG trafficking inside infected HEMC-1 cells ($5 \times 10^4$), we infected them with HCMV at an MOI of 5 for 48 h and then incubated them with 250 µg/ml IgG for 1 h at 37 °C. To visualize IgG trafficking inside US11$^+$ cells, HeLa$^{FcRn+US11}$ and HeLa$^{FcRn}$ cells were also incubated with 250 µg/ml IgG for 1 h at 37 °C. After washing, cells were incubated for an additional 1 h in complete medium containing 5% FBS with ultra-low IgG, and then fixed and stained by immunofluorescence for co-localization of IgG with the early endosomal marker EEA1 or lysosomal marker LAMP-1. For Pearson's correlation coefficient measurement, ten microscopic fields, each of which contained at least ten cells, were measured for correlation coefficiency rate.

**Statistics**. The differences between groups were tested by an unpaired two-tailed Student's $t$-test with a significance level of 0.05. Data are expressed as mean ± SEM.

**Reporting summary**. Further information on research design is available in the Nature Research Reporting Summary linked to this article.

## Data availability

Sequence data that support this article are available in Genbank under the primary accession code MK647994. The authors declare that all other relevant data supporting the findings of this study are available within the Article and its Supplementary Information files, or from the corresponding author on reasonable request. A Reporting Summary for this Article is available as a Supplementary Information file. The source data for figures are provided in the Source Data file.

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

## Acknowledgements

We are most grateful for discussions from Dr. David Mosser, Dr. Georgiy Belov, Dr. Yanjin Zhang, and Dr. Siba Samal. We are grateful to Dr. Najib M. El-Sayed, Dr. Fenyong Liu, and Dr. Hua Zhu for supplying us with cell lines and virus. We acknowledge the receipt of FcRn or pTFR1-GFP protein expression plasmids and cell lines from Dr. Richard S. Blumberg and Dr. Gary Banker. This work was in part supported by National Institutes of Health (NIH) grants AI130712 (X.Z.), AI102680 (C.D.P. and X.Z.), the Universality of Maryland MAES competitive grant (X.Z.), fellowships from the NIH T32 grant AI125186 (S.P.-O.), the Chinese Scholar Council (J.T. and X.W.), and the American Association of Immunologists (AAI) Careers in Immunology Fellowship (S.P. and X.Z.). J.I.C. is supported by the intramural research program of the National Institute of Allergy and Infectious Diseases.

## Author contributions

X.L. and X.Z. designed and performed experiments, analyzed data, and wrote the paper. S.P., I. Z., J.T., W. L., X.W., and S.P.-O. conducted experiments. J.I.C., I.Z., and C.D.P. provided the reagents and materials, performed data analysis, provided editorial suggestions. X.Z. directed the project.

## Additional information

**Competing interests:** X.Z. and X.L. declare that they have filed a provisional patent application together with the University of Maryland (LS-2018-054) for the application of blocking FcRn function as a means to therapy or to enhance vaccine design. The other authors declare no competing interests.

