## [Peer Review File · Nature Communications]

Reviewers' comments:

Reviewer #1 (Remarks to the Author):

Liu et al provide an interesting study demonstrating that human cytomegalovirus US11 protein binds to and causes degradation of the neonatal Fc receptor (FcRn or FCGRT). Specifically, US11 retains immature FcRn in the ER, preventing its assembly with $\beta 2m$ and trafficking to the endosome. A consequence of this interaction is recruitment of several ligases that are used in the US11-mediated dislocation and degradation of MHC class I molecules, namely Derlin-1, TMEM129 and UBE2J2. These result in the cytoplasmic dislocation and proteasomal degradation of FcRn. The authors show, in both colonic and placental epithelial cell lines that the consequences of this interaction include diminished IgG transcytosis, and intracellular IgG degradation.

The study is very well written and clear, and in general the experiments provide comprehensive, clear data supporting their conclusions. The study would be of interest to those in the HCMV field and beyond. I have few concerns (detailed below) and believe that with suitable modification this manuscript would be a good candidate for Nature Communications.

Major comments:

1. The authors examine a range of HCMV proteins for interaction with FcRn (page 17) as a screen. This data is not shown – it should be, at least as supplemental information.

2. Further details of their new HCMV clinical strain are required. How was this obtained – from urine or blood? Virus after how many passages were used in the various experiments? Was it cloned into a BAC? Was it sequenced? Have they named this strain for future reference in the literature? They use this strain at high MOI in cells that require an endotheliotropic virus, so maintenance of low passage stocks would be important. Please also provide details of the institution review board ethics approval (number, date).

3. Cell types usually used for HCMV infections in the majority of the previous literature have most frequently included human fibroblast lines (e.g. MRC5s or HFFFs), primary endothelial cells or differentiated THP-1s. For the majority of their studies, the authors employ two distinct cell types, namely Caco-2 cells (a colon epithelial cell-derived cell line) and HeLas. Caco-2 cells were established as a model by Jay Nelson's group in 1999, however have not since widely been used. Furthermore, infection is differentiation-state dependent, and infection does not spread from cell to cell. Similarly there is a block to efficient replication in HeLas. This having been said, the Caco-2 cells clearly are more appropriate to address questions of IgG transcytosis. The authors demonstrate an interaction between US11 and FcRn in THP-1s and HMEC (endothelial line), and show reduction in FcRn during HCMV infection in the same cells. However, the authors should repeat at least their key finding (direct demonstration of US11 effect on FcRn during HCMV infection, preferably via comparison between a US11 deletion virus and wild-type virus but alternatively using efficient US11 knockdown using si or shRNA) in one of the cell lines typically used in HCMV research that expresses FcRn. For example, in a paper cited by the authors, Fielding et al 2017 eLife, plasma membrane and whole cell proteins regulated by the HCMV US12-21 gene region were measured in HFFF. HCMV downregulated cell surface FcRn in HFFF, suggesting these would be a tractable and more typical model of infection. They could infect these cells with their clinical strain, or AD169 (which expresses US11), or another commonly used strain. I note in the discussion that the authors suggest that a US11 deletion virus was not useful due to the effect of other (unspecified) viral proteins that interact with FcRn. However, their results from siRNA experiments suggest a deletion virus should validate the phenotype discussed above.

4. The authors should discuss precise details of THP-1 infections in their methods section. THP-1 cells would usually be differentiated prior to full-cycle replication with HCMV – did this occur? What was the differentiation state of the Caco-2 cells?

5. In the reference Fielding et al 2017, HCMV US20 downregulated FcRn (FCGRT). It is clearly possible that more than one HCMV gene regulates FcRn (as the authors suggest in their discussion), or that different genes are most important in different cell types. To address this

potential criticism, possibly as part of the experiment suggested in point (3) above, the authors should compare US11 and US20 effect on FcRn in both their Caco-2 model and a more typical cellular model of HCMV infection (e.g. human fibroblasts), either using si or sh RNA to knock down US11 and US20, or more preferably using deletion viruses for these two HCMV genes (which are available and published).

Minor comments

1. Page 4 'U3' should read 'US3'
2. Is the decrease in colocalisation of FcRn and EEA1 on page 19 due to degradation of FcRn in US11-expressing cells? How was decreased expression of FcRn controlled for?

Reviewer #2 (Remarks to the Author):

The manuscript entitled "Human cytomegalovirus evades IgG antibody-mediated immunity through endoplasmic reticulum-associated degradation of the neonatal Fc receptor (FcRn) for IgG" by Liu et al describes the human cytomegalovirus (CMV) US11-mediated destruction of FcRn in cells expressing US11 and virus infected cells. The innovative aspect of the manuscript is that FcRn is the latest protein down-regulated by HCMV US11 and likely contributes to viral evasion strategy of the virus. In addition, the authors perform several experiments to address the possible consequences of preventing FcRn from being expressed on the cell surface. However, the association of ERAD components with FcRn in a US11 dependent manner is consistent with published findings. Also, there are issues with the manuscript that are outlined below.

- 1) The initial association studies demonstrate an interaction of US11 with FcRn. Yet, US11 mediates the destruction of FcRn in latter Figures. In addition, equivalent levels of FcRn are observed in cells that express US11. These results would have to be rationalized because it suggests that FcRn is not being degraded in these studies.
- 2) Many of the experiments lack an IP control. For example, the immunoprecipitation of US11 reveals an association with FcRn in many experiments. However, the blot should be probed with a different membrane protein to ensure the specificity of the immunoprecipitation. This issue occurs throughout the manuscript.
- 3) The US11-mediate down regulation of FcRn (Figure 3) should include the analysis of MHC class I to validate that US11 continues to be active in these cells.
- 4) The quantification of polypeptides in the kinetics studies should use FcRn levels over tubulin as a better representative of the data.
- 5) Figure 6 should include a blot probing for ubiquitin levels in total cell lysates to better evaluate the amount of ubiquitin species associated with FcRn.
- 6) The rationale for Figure 7 is not clear and does not add to the story.

Reviewer #3 (Remarks to the Author):

In their manuscript entitled „Human cytomegalovirus evades IgG antibody-mediated immunity through endoplasmic reticulum-associated degradation of the neonatal Fc receptor (FcRn) for IgG“, Liu et al. investigate the interaction between US11, a CMV-encoded protein that is involved in shuttling MHC class I heavy chains from the endoplasmic reticulum (ER) to the cytosol for proteasomal degradation, and the neonatal Fc receptor (FcRn).

FcRn is important for transporting IgG across cellular membranes including the human placenta. A wealth of data are available that show an interference of CMV with the cellular immune response, however, few data are published that highlight an interaction between HCMV and the humoral arm of the immune system. Here, Liu et al. demonstrate novel data in that infection by HCMV results in degradation of FcRn and that US11, at least in part, is responsible for this process. These data

should be of interest to others in the community and beyond.

Major criticism.

1. The authors employed the AD169 strain but apparently also isolated CMV from a patient. Throughout the manuscript, it is stated that cells were infected with CMV but no information is given if cells were infected with AD169 or the clinical isolate. What is more, it is imperative that this clinical isolate is characterized before it is used in the study, i.e., sequence information should be provided and deposited in Genbank. By using the separation method described, can the authors rest assured that the isolated virus lacks any cellular components?
2. At the beginning of their study, the authors probed various HCMV proteins for interactions with FcRn. These data should be shown to understand why they selected US11 for further analyses.
3. Figures 1C and 2 A/C. As a control, cells need to be transfected with US11 alone. Also, a combination of different secondary antibodies should be used since the spectra of Alexa-Fluor 488 and -555 might overlap. Is the expression of US11 comparable between the different cell types used here?
4. Figure 3G/K. AD169 efficiently replicates only in fibroblasts. How efficiently do the viruses employed in this study replicate in Caco-2 cells?
5. Figure 3H. How do the authors explain the difference between HCMV + si US11 and mock at 240 min?
6. The authors examine the interaction between US11 and FcRn or other components and demonstrate interactions, however, these experiments are largely based on co-immunoprecipitations. It would be useful to show data to complement the findings, e.g., by flow cytometry. This might be especially important for demonstrating the interaction between the extracellular domains of US11 and FcRn (Fig. S4) since GST binding can be non-specific.
7. All data were generated in cell lines and the results are based on transfection experiments. Would there be a possibility to utilize primary cells/trophoblasts?
8. Towards the end of their manuscript, the authors discuss the possibility of generating a US11 knockout virus and acknowledge that other HCMV proteins might be involved in targeting FcRn although FcRn „is relatively non-polymorphic“ (page 34). Instead of using siRNA (or rather to confirm the results), a US11 knockout virus should be used that is available (Schempp S et al., Virus Research 155 (2011), 446-454) or that can be generated from a newly constructed BAC clone with a repaired US2-6 region (Laib Sampaio K et al., BioTechniques 63 (2017), 205-214). Further, the authors need to show which other HCMV proteins might be involved in downregulating FcRn.
9. What are the implications for primary infection and/or reactivation in pregnant women, if any? Is there a role for FcRn in superinfection (secondary HCMV infection) when a pregnant woman has already been infected by CMV?

Minor points.

Page 4, introduction, second paragraph: „HCMV has been extremely successful in infecting humans...“. While it is true that CMV is a master of immune evasion, many herpesviruses have been successful in infecting humans and seroprevalence is higher among, e.g., HSV and VZV. With HCMV, there are geographical regions where only approximately 50% of the population are infected. The beginning of the sentence might be reworded.

Page 4, line 8 from the bottom: ...the US3 protein...

Page 6, bottom paragraph: Human CD34-positive cells (hematopoietic stem cells) should be included in the list of cell types that are infected by HCMV. Is FcRn expressed at similar levels in these cell types?

Page 9, bottom paragraph: The introductory sentence is incomplete.

Page 43, figure legends. First line: The sentence is incomplete.

Page 52, figure 9. What determines the portion of beta2-microglobulin-free FcRn HC molecules that associate with US11 in the ER?

Reviewers' comments:

Reviewer #1 (Remarks to the Author):

Liu et al provide an interesting study demonstrating that human cytomegalovirus US11 protein binds to and causes degradation of the neonatal Fc receptor (FcRn or FCGRT). Specifically, US11 retains immature FcRn in the ER, preventing its assembly with β 2m and trafficking to the endosome. A consequence of this interaction is recruitment of several ligases that are used in the US11-mediated dislocation and degradation of MHC class I molecules, namely Derlin-1, TMEM129 and Ube2J2. These result in the cytoplasmic dislocation and proteasomal degradation of FcRn. The authors show, in both colonic and placental epithelial cell lines that the consequences of this interaction include diminished IgG transcytosis, and intracellular IgG degradation.

Fig.1. A. Constructions of plasmid vectors encoding HCMV genes involved in immune evasion.

The study is very well written and clear, and in general the experiments provide comprehensive, clear data supporting their conclusions. The study would be of interest to those in the HCMV field and beyond. I have few concerns (detailed below) and believe that with suitable modification this manuscript would be a good candidate for Nature Communications.

Major comments:

1. *The authors examine a range of HCMV proteins for interaction with FcRn (page 17) as a screen. This data is not shown – it should be, at least as supplemental information.*

[redacted]

Responses: Several HCMV proteins have been previously shown to interact with MHC class I. We have cloned most of these HCMV US and UL genes from HCMV AD169 DNA by PCR using primers that either incorporated influenza virus hemagglutinin (HA) epitope tag at the C-terminus of the proteins US2, US3, US6, US10, US11, UL16, and UL18 (Fig. 1A). These plasmids were sequenced for verifying PCR amplification and the correct ORF of cloning. In addition, we have established a stable HeLa cell expressing both FcRn HC and β 2m. The N-terminus of the FcRn protein is tagged with a FLAG epitope; the addition of the FLAG tag at the N-terminus

of the FcRn protein does not affect FcRn function. To screen the interaction of FcRn and HCMV proteins, we transfected these plasmids into the stable HeLa-FcRn cells. All these proteins were expressed, as assessed by HA-specific mAb in either Western blotting the cell lysates or by immunofluorescence (Fig. 1B). We found [redacted], US11, and [redacted] interacted with human FcRn.

To further verify whether HCMV proteins ([redacted]) interact with FcRn, the immunoprecipitation and Western blot coupling experiments were performed. In our studies, we showed HCMV [redacted], US11, [redacted] ([redacted]) proteins interacted with FcRn in HeLa cells. As shown in [redacted], anti-HA Ab co-immunoprecipitated FcRn heavy chain ([redacted]) and anti-FLAG antibody co-immunoprecipitated the [redacted], US11, and [redacted] protein ([redacted]). In the submitted manuscript, we focus on HCMV US11 protein to understand the how US11 causes FcRn degradation and affects FcRn function.

Our parallel studies did not find evidence either [redacted] protein mediated FcRn degradation. We are currently performing studies to detail how [redacted] protein blocks FcRn function. A new manuscript will report these results and we will cite this submitted US11 manuscript as a reference. Hence, we prefer to not disclose the detailed information about the interaction of FcRn with either [redacted] protein. We do mention this in a sentence at the top of page 6 as unpublished data in revised manuscript. In addition, this US11 manuscript already contains large volume of information and many Figures.

2. Further details of their new HCMV clinical strain are required. How was this obtained – from urine or blood? Virus after how many passages were used in the various experiments? Was it cloned into a BAC? Was it sequenced? Have they named this strain for future reference in the literature? They use this strain at high MOI in cells that require an endotheliotropic virus, so maintenance of low passage stocks would be important. Please also provide details of the institution review board ethics approval (number, date).

Responses: We have provided the information about new HCMV clinical strain in the revised Methods section (bottom of page 25). The virus, designated for CMV Bethesda BAL, was isolated from bronchoalveolar lavage fluid from a patient at the National Institutes of Health who signed consent for a protocol (01-I-0161) that was approved by the Internal Research Board of the National Institute of Allergy and Infectious Diseases. The virus was grown in MRC-5 cells and was passaged less than 5 times in MRC-5 cells before use in our experiment. It was not cloned into a BAC and has not been completely sequenced.

3. Cell types usually used for HCMV infections in the majority of the previous literature have most frequently included human fibroblast lines (e.g. MRC5s or HFFFs), primary endothelial cells or differentiated THP-1s. For the majority of their studies, the authors employ two distinct cell types, namely Caco-2 cells (a colon epithelial cell-derived cell line) and HeLas. Caco-2 cells were established as a model by Jay Nelson's group in 1999, however have not since widely been used. Furthermore, infection is differentiation-state dependent, and infection does not spread from cell to cell. Similarly, there is a block to efficient replication in HeLas. This having been said, the Caco-2 cells clearly are more appropriate to address questions of IgG transcytosis. The authors demonstrate an interaction between US11 and FcRn in THP-1s and HMEC (endothelial line), and show reduction in FcRn during HCMV infection in the same cells. However, the authors should repeat at least their key finding (direct demonstration of US11 effect on FcRn during HCMV infection, preferably via comparison between a US11 deletion virus and wild-type virus but alternatively using efficient US11 knockdown using si or shRNA) in one of the cell lines typically used in HCMV research that expresses FcRn. For example, in a paper cited by the authors, Fielding et al 2017 eLife, plasma membrane and whole cell proteins regulated by the HCMV US12-21 gene region were measured in HFFF. HCMV downregulated cell surface FcRn in HFFF, suggesting these would be a tractable and more typical model of infection. They could infect these cells with their clinical strain, or AD169 (which ex-

presses US11), or another commonly used strain. I note in the discussion that the authors suggest that a US11 deletion virus was not useful due to the effect of other (unspecified) viral proteins that interact with FcRn. However, their results from siRNA experiments suggest a deletion virus should validate the phenotype discussed above.

Responses: We have now included different cell lines (HeLa, Caco, THP-1, HMEC-1) and primary HUVEC cells in the revised manuscript. These new data are now included in Figures 1F, 1G, 3G, 3I, 3K, and Supplemental Figures 2,3G, 3H, and 5 of the revised manuscript. We also determined whether human FcRn is expressed in MRC-5 cells and human foreskin fibroblasts (HFF) in a Western blot. Interestingly, we were unable to detect FcRn protein expression in both MRC-5 and HFF cell lines (Fig. 3), although some level of mRNA was detected by RT-PCR. It is possible, HCMV infection can induce FcRn expression in these fibroblast cell lines, similarly, we failed to detect FcRn protein expression in virally infected MRC-5 and HFF cell lines (Fig. 3). Human primary endothelial cell line HUVEC was used as a positive control. Hence, we decided we will not further analyze FcRn expression in MRC-5 and HFF cell lines in this study.

Our FcRn protein result is different from the results in Fielding et al 2017 eLife, which HCMV downregulated cell surface or intracellular FcRn in HFF. This discrepancy may be caused by protein detection method. Our experience is that the data from microarray, mass spectrometry, or RT-PCR analysis need to be verified by Western blot or immunofluorescence staining.

4. The authors should discuss precise details of THP-1 infections in their methods section. THP-1 cells would usually be differentiated prior to full-cycle replication with HCMV – did this occur? What was the differentiation state of the Caco-2 cells?

Responses: We have provided more detailed information about THP-1 cells infected with HCMV in the revised Methods, bottom of page 24. To differentiate THP-1 cells into macrophages, cells were treated with 50 ng/ml Phorbol-12-myristate-13-acetate (PMA) for 48 hrs. We found HCMV US11 interacted with FcRn in THP-1 cells in either monocytic or macrophage state (Fig. 4).

5. In the reference Fielding et al 2017, HCMV US20 downregulated FcRn (FCGRT). It is clearly possible that more than one HCMV gene regulates FcRn (as the authors suggest in their discussion), or that different genes are most important in different cell types. To address this potential criticism, possibly as part of the experiment suggested in point (3) above, the authors should compare US11 and US20 effect on FcRn in both their Caco-2 model and a more typical

Fig 3. Detection of human FcRn expression in fibroblasts. Cell lysate (20 μg) from HFF, MRC-5, and HUVEC cells were blotted with human FcRn-specific Abs. HUVEC was used as a positive control

Fig 4. HCMV US11 interacts with FcRn in PMA-treated THP-1 cells. THP-1 cells were treated with 50 ng/ml PMA for 48 hrs, the differentiated THP-1 cells were mock-infected or infected with HCMV at MOI of 5. 24 later, the cell lysates were immunoprecipitated by mouse anti-US11 Abs (Panel A) or rabbit anti-human FcRn Abs (Panel B). The immunoprecipitates were subjected to Western blotting with anti-human or US11 Abs as indicated. Cell lysate from each sample with equal amounts of total protein (input, 20 μg) were blotted with the indicated Abs.

cellular model of HCMV infection (e.g. human fibroblasts), either using si or sh RNA to knock down US11 and US20, or more preferably using deletion viruses for these two HCMV genes (which are available and published).

Responses: Thanks for the reviewer to point out this issue. We have carefully read the paper published by Fielding et al. (2017). The major theme of this paper is to show that US20 regulates intracellular endo-lysosomal vesicular transport of B7-H6, finally causing B7-H6 degradation. Although this paper provided the mass spectrometric evidence that HCMV US20 downregulated FcRn (FCGRT) and other proteins (25 surface proteins or 16 intracellular proteins) in Figure 1. There are no follow-up studies to verify whether US20 interacts with FcRn or cause FcRn degradation.

First, we cloned US20 gene into the plasmid pEF6 by using a primer pair. The plasmid was sequenced to verify the amplification fidelity and cloning. To show whether US20 interacts with FcRn, HeLa^{FcRn} or HeLa cells were transfected with plasmids encoding HA-tagged US20 cDNA. HeLa^{US11+FcRn} cells or HeLa cells were used as a positive or negative control. Cells lysed with CHAPS buffer were used for immunoprecipitation with either anti-FLAG (for FcRn) (Fig. 5A) or anti-HA (for US20 or US11) (Fig. 5B) mAb. Using Western blotting with anti-HA and anti-FLAG Ab, we show that the anti-FLAG Ab for FcRn failed to coimmunoprecipitate US20 protein (Fig. 5A, lane 2) and the anti-HA Ab for US20 did not pull down FcRn heavy chain (HC) (Fig. 5B, lane 2). As a position control, we showed that the anti-FLAG Ab coimmunoprecipitated US11 protein (Fig. 5A, lane 1) and the anti-HA Ab coimmunoprecipitated FcRn heavy chain (HC) (Fig. 5B, lane 1). Therefore, we conclude

Fig 5. US20 didn't interact with FcRn. pEF6-US20-HA plasmids carrying HA-tagged US20 were transfected into HeLa^{FcRn} or HeLa cells. 48 later, the cell lysates were immunoprecipitated by mAb anti-FLAG for FcRn (Panel A) or anti-HA for either US20 or US11 (panel B). The immunoprecipitates were subjected to Western blotting with anti-FLAG or HA mAb as indicated. Cell lysate from each sample with equal amounts of total protein (input, 20 µg) were blotted with the indicated Abs.

Fig 6. HeLa^{FcRn} cells were transfected with US11 (A) or US20 (B) plasmids. A stable HeLa^{FcRn+US11} (A) and HeLa^{FcRn+US20} (D) cell lines were then treated with CHX (100 µg/ml) for the indicated time. The cells were lysed after CHX treatment, protein levels were measured, and Western blotting-ECL was performed using FLAG (FcRn), HA (US11 or US20), and tubulin antibodies.

Fig 7. The level of remaining FcRn at different time points was quantified as the percentage of the β-tubulin level.

that US20 didn't interact with FcRn.

Although HCMV US20 didn't interact with FcRn, it is possible that US20 causes FcRn degradation in an indirect manner. To specifically monitor the rate of FcRn HC degradation, we again performed a quantitative cycloheximide (CHX) chase assay. HeLa^{FcRn + US11} (Fig. 6A) and HeLa^{FcRn + US20} (Fig. 6B) cells were treated with CHX (100 µg/ml) and FcRn protein intensity was detected in Western blot for the indicated times. In HeLa^{FcRn + US11} cells, the expression of US11 induced a significant and time-dependent decrease in FcRn protein levels (Fig. 6A) in comparison with that of HeLa^{FcRn + US20} cells (Fig. 6B). Hence, we did not detect a significant change in the levels of FcRn expression in HeLa^{FcRn + US20} cells (Fig. 3F), suggesting that US20 has no a direct effect on FcRn stability in this degradation assay.

In the paper published by Fielding et al. (2017), US20 affected the expression levels of 25 surface proteins or 16 intracellular proteins, suggesting a non-specific regulation. It remains unclear how US20 protein affects different proteins, more work needs to be done in the future. Overall, we prefer we will not discuss or include US20 finding in our manuscript.

Minor comments

1. Page 4 'U3' should read 'US3'

Response: We have corrected this careless mistake in the revised manuscript.

2. Is the decrease in colocalisation of FcRn and EEA1 on page 19 due to degradation of FcRn in US11-expressing cells? How was decreased expression of FcRn controlled for?

Responses: The reviewer raises an excellent point. The decrease in colocalization of FcRn and EEA1 in endosome is finally due to US11-mediated FcRn sequestration in the ER and FcRn degradation in proteasome in US11-expressing cells. The reduction of colocalization between FcRn and EEA1 is unlikely due to the decreased expression of FcRn in US11-expressing cells because the same HeLa cell line stably expressing FcRn was used for transfecting US11 plasmid and for control cells to monitor the level of FcRn expression. To reduce the confusion, we have emphasized this point in the revised manuscript (top of page 8).

Reviewer #2 (Remarks to the Author):

The manuscript entitled “Human cytomegalovirus evades IgG antibody-mediated immunity through endoplasmic reticulum-associated degradation of the neonatal Fc receptor (FcRn) for IgG” by Liu et al describes the human cytomegalovirus (CMV) US11-mediated destruction of FcRn in cells expressing US11 and virus infected cells. The innovative aspect of the manuscript is that FcRn is the latest protein down-regulated by HCMV US11 and likely contributes to viral evasion strategy of the virus. In addition, the authors perform several experiments to address the possible consequences of preventing FcRn from being expressed on the cell surface. However, the association of ERAD components with FcRn in a US11 dependent manner is consistent with published findings. Also, there are issues with the manuscript that are outlined below.

1) The initial association studies demonstrate an interaction of US11 with FcRn. Yet, US11 mediates the destruction of FcRn in latter Figures. In addition, equivalent levels of FcRn are observed in cells that express US11. These results would have to be rationalized because it suggests that FcRn is not being degraded in these studies.

Responses: The reviewer raised an excellent point. FcRn is constantly synthesized in cells. In spite of this, we can easily detect FcRn expression level was lower in HCMV-infected human cells, such as THP-1, HMEC-1, HUVEC (Fig. 8), in comparison with that of uninfected cells. In transfected HeLa cells, US11-mediated degradation was more obvious in CHX-treated HeLa^{FcRn + US11} cells because a strong CMV promoter in pCDNA3 plasmid was used to drive FcRn expression in HeLa cells.

2) Many of the experiments lack an IP control. For example, the immunoprecipitation of US11 reveals an association with FcRn in many experiments. However, the blot should be probed with a different membrane protein to ensure the specificity of the immunoprecipitation. This issue occurs throughout the manu-

Fig 8. F+G. US11 interacts with FcRn in HCMV-infected human primary umbilical vein endothelial cells (HUVEC). The HUVEC were infected with HCMV virus at a MOI of 5. At day 2 p.i., the cell lysates from infected or mock-infected HUVEC were immunoprecipitated with anti-US11 Ab (F) or anti-FcRn Ab (G). The immunoprecipitates were subjected to 12% SDS-PAGE electrophoresis under reducing conditions, then transferred to a nitrocellulose membrane for Western blotting with anti-FcRn or US11 Ab as indicated. The cell lysates (20 µg) were blotted as controls. Immunoblots (IB) were developed with ECL.

Fig. 9. US11 does not interact with endogenous transferrin receptor (TfR) and FcRn does not interact with the HCMV US2. A. The cell lysates from HeLa^{FcRn+US11} (lane 1), HeLa^{FcRn} (lane 2), HeLa^{US11} (lane 3), and HeLa control (lane 4) were immunoprecipitated by mAb anti-HA for US11. The immunoprecipitates were subjected to 12% SDS-PAGE electrophoresis under reducing conditions, then transferred to a nitrocellulose membrane for Western blotting with anti-TfR, anti-FLAG (FcRn), or HA (US11) mAb as indicated. Immunoblots (IB) were developed with ECL. The 50 µg cell lysates (input) were blotted with the indicated Abs. The location of the TfR, FcRn HC or US2 is indicated by an arrow.

script.

Responses: We appreciate the reviewer's meticulousness and honest comment, however, we somewhat disagree with this critique. To show the specificity, we co-expressed FcRn with US2, and US11 with HFE, which possesses structural similarity to FcRn, at similar levels in HeLa cells. We failed to detect any interaction between FcRn and US2 or between HEF and US11 in a reciprocal immunoprecipitation experiment, indicating a high degree of interaction specificity between FcRn and US11. To respect the reviewer's comment, we also immunoprecipitated US11 and endogenous transferrin receptor 1 in HeLa cells; US11 failed to pull down TfR1 (Fig. 9, the Supplementary Fig. 1 in the revised manuscript), suggesting its specificity. We also found US11 did not interact with mouse FcRn, showing species specificity (data not shown).

3) The US11-mediated down regulation of FcRn (Figure 3) should include the analysis of MHC class I to validate that US11 continues to be active in these cells.

Responses: The reviewer raised an excellent point. We have already performed such experiments for validating US11-mediated MHC class I degradation or ubiquitination assays (Fig. 10, the Supplementary Fig. 11 & 12 in the revised manuscript). These experiments have been included and discussed in the revised manuscript.

4) The quantification of polypeptides in the kinetics studies should use FcRn levels over tubulin as a better representative of the data.

Responses: We appreciate the reviewer's point and agree with the reviewer. In the kinetics studies, we have quantified FcRn levels over tubulin inter-

Fig 11. The level of remaining endogenous FcRn (at different time points) was quantified as the percentage of β-tubulin content. Each experiment was carried out three times.

nal control (*Fig. 11*). The scientific conclusions were not affected by the modifications. The corresponding figures have been replaced with the modified Figures in the revised manuscript (*Figures 3D, 3F, 3H, 3J, 4E, 5C, and 6I in the new manuscript*).

5) *Figure 6 should include a blot probing for ubiquitin levels in total cell lysates to better evaluate the amount of ubiquitin species associated with FcRn.*

Responses: We agree with the reviewer. We have performed a blot probing the ubiquitin levels in total cell lysates (*Fig. 12*), as it showed a smear background. We have included this new information in the revised manuscript (*Figures 6A, 6B and Supplemental Figure 7B*).

6) *The rationale for Figure 7 is not clear and does add to the story.*

Responses: Thanks for raising this point. The aim of the Figure 7 is to examine whether HCMV infection or US11 alone affects FcRn-mediated IgG transcytosis across polarized epithelial cells. IgG transcytosis across the polarized epithelial cells represents an hallmark function of FcRn. It is important to examine the effect of US11 on IgG transport by FcRn. We have provided the rationale in the corresponding section of the revised manuscript.

Fig 12. FcRn is ubiquitinated in the presence of US11 and MG132. HeLa^{FcRn} (A), HeLa^{HFE} (B) cells were transfected with or without US11 plasmids for 48 hr, and cells were treated with proteasome inhibitor MG132 (50 µM) for 2 hr, as indicated. Cell lysates (0.5 mg) were immunoprecipitated with mAb anti-FLAG for FcRn (A) or HFE (B). Immunoprecipitates were subjected to the electrophoresis and immunoblotting analysis to detect ubiquitin and the target proteins FcRn, HFE, US11, or β-tubulin with corresponding Abs, as indicated. The ubiquitins in the cell lysates (20 µg, A+B) were blotted as an intern control.

Reviewer #3 (Remarks to the Author):

In their manuscript entitled, human cytomegalovirus evades IgG antibody-mediated immunity through endoplasmic reticulum-associated degradation of the neonatal Fc receptor (FcRn) for IgG^a, Liu et al. investigate the interaction between US11, a CMV-encoded protein that is involved in shuttling MHC class I heavy chains from the endoplasmic reticulum (ER) to the cytosol for proteasomal degradation, and the neonatal Fc receptor (FcRn).

FcRn is important for transporting IgG across cellular membranes including the human placenta. A wealth of data are available that show an interference of CMV with the cellular immune response, however, few data are published that highlight an interaction between HCMV and the humoral arm of the immune system. Here, Liu et al. demonstrate novel data in that infection by HCMV results in degradation of FcRn and that US11, at least in part, is responsible for this process. These data should be of interest to others in the community and beyond.

Major criticism.

1. The authors employed the AD169 strain but apparently also isolated CMV from a patient. Throughout the manuscript, it is stated that cells were infected with CMV but no information is given if cells were infected with AD169 or the clinical isolate. What is more, it is imperative that this clinical isolate is characterized before it is used in the study, i.e., sequence information should be provided and deposited in Genbank. By using the separation method described, can the authors rest assured that the isolated virus lacks any cellular components?

Responses: See response to reviewer 1, comment 2. We have not sequenced HCMV and have published HCMV strains extensively. Since it was not molecularly cloned it should not need to be sequenced. It grows epithelial cells and fibroblasts; it should not lack any critical genes. We recently cloned US20 from the clinical strain HCMV, the sequence is matched with the US20 sequence deposited in Genbank.

2. At the beginning of their study, the authors probed various HCMV proteins for interactions with FcRn. These data should be shown to understand why they selected US11 for further analyses.

Responses: The Reviewer 1 also raised the same question. This question has been well addressed in response to Reviewer 1's Question 1. We have identified HCMV [redacted], US11, and [redacted] interacted with FcRn. However, [redacted] did not mediate FcRn degradation. We are further studying the impact of [redacted] on FcRn biology and functions with the NIH grant support. New manuscripts will be submitted in the future, this US11 manuscript will be cited as a reference if accepted by the Nature Communications.

3. *Figures 1C and 2 A/C. As a control, cells need to be transfected with US11 alone. Also, a combination of different secondary antibodies should be used since the spectra of Alexa-Fluor 488 and -555 might overlap. Is the expression of US11 comparable between the different cell types used*

Fig 13. Colocalization of FcRn and US11 in HeLa^{FcRn+US11} cells. HeLa^{FcRn} cells or HeLa^{US11} cells were used as a control. Cells grown on coverslips were fixed with 4% paraformaldehyde and permeabilized in 0.2% Triton X-100. Subsequently, the cells were incubated with affinity-purified anti-FLAG (FcRn) or anti-HA (US11) specific mAb, followed by Alexa Fluor 488- or 555-conjugated IgG. Puncta that appear yellow in the merged images (right panel) indicate colocalization of FcRn with US11 protein. The nuclei were stained with DAPI (blue). Scale bar represents 10 μ m.

here?

Responses: We have performed such an experiment (Fig. 13) and included the Figure in the revised manuscript (Figure 1C middle panel of the revised manuscript).

4. Figure 3G/K. AD169 efficiently replicates only in fibroblasts. How efficiently do the viruses employed in this study replicate in Caco-2 cells?

Response: The clinical strain HCMV was used. To reduce the confusion, we have highlighted this point in the revised manuscript.

5. Figure 3H. How do the authors explain the difference between HCMV + si US11 and mock at 240 min?

Responses: The reviewer raised an excellent point. We found that FcRn degradation was significantly reduced in US11 siRNA-treated HUVEC or Caco-2, although β_2m levels were unaffected in HUVEC cells or Caco-2 cells. We also noticed that FcRn level at 240 min post chase was moderately restored by US11 siRNA in viral infected cells in comparison with mock-infected cells. It is likely that this result was associated with the incomplete blocking of US11 expression by US11 siRNA, which was shown in US11 blot. We believe that this incomplete knockdown of US11 was likely associated with the transfection efficiency of US11 siRNA. It is difficult to reach 100% transfection efficiency by delivering US11 siRNA into all cells. We have emphasized this point in the revised manuscript (page 10, bottom).

6. The authors examine the interaction between US11 and FcRn or other components and demonstrate interactions, however, these experiments are largely based on co-immunoprecipitations. It would be useful to show data to complement the findings, e.g., by flow cytometry. This might be especially important for demonstrating the interaction between the extracellular domains of US11 and FcRn (Fig. S4) since GST binding can be non-specific.

Responses: FcRn in the absence of β_2m association is not able to exit the ER and traffics to cell surface; US11 interacted with β_2m -free FcRn but to FcRn/ β_2m complex. Therefore, it is difficult to use flow cytometry to examine the interaction of the extracellular domains of US11 and FcRn. We used flow cytometry to show HCMV or US11 downregulations of cell surface or intracellular FcRn expression level by staining with FcRn-specific antibody.

We are confident about specific pull-down of FcRn using the purified GST-US11 protein from the lysates of HeLa^{FcRn} but not HeLa cells. In addition, GST alone did not pull-down any proteins from either lysates of the HeLa^{FcRn} or HeLa cells.

7. All data were generated in cell lines and the results are based on transfection experiments. Would there be a possibility to utilize primary cells/trophoblasts?

Responses: To respect the re-

Fig. 14. US11 interacts with FcRn in HCMV-infected human primary umbilical vein endothelial cells (HUVEC). The HUVEC were infected with HCMV virus at a MOI of 5. At day 2 p.i., the cell lysates from infected or mock-infected HUVEC were immunoprecipitated with anti-US11 Ab (F) or anti-FcRn Ab (G). The immunoprecipitates were subjected to 12% SDS-PAGE electrophoresis under reducing conditions, then transferred to a nitrocellulose membrane for Western blotting with anti-FcRn or US11 Ab as indicated. The cell lysates (20 μ g) were blotted as controls. Immunoblots (IB) were developed with ECL.

viewer's comment, we have obtained primary human umbilical vein endothelial cells (HUVEC) from ATCC. The HUVEC was infected with clinical strain HCMV at a MOI of 5. At day 2 post-infection (p.i.), cell lysates from infected or mock-infected cells were immunoprecipitated with anti-US11 Ab (Fig. 14F) or anti-FcRn Ab (Fig. 14G) in HUVEC cells. We then determined that anti-US11 Ab co-immunoprecipitated with FcRn HC in infected cells (Fig. 14F). Similarly, anti-FcRn Ab was also found to co-immunoprecipitate with US11 protein in infected cells (Fig. 14G). We have included this new information in the revised manuscript (*new Figures 1F, 1G, 3G, 3I, 3K, and Supplemental Figures 2, 3G, 3H, and 5*). .

8. *Towards the end of their manuscript, the authors discuss the possibility of generating a US11 knockout virus and acknowledge that other HCMV proteins might be involved in targeting FcRn although FcRn is relatively non-polymorphic“ (page 34). Instead of using siRNA (or rather to confirm the results), a US11 knockout virus should be used that is available (Schempp S et al., Virus Research 155 (2011), 446-454) or that can be generated from a newly constructed BAC clone with a repaired US2-6 region (Laib Sampaio K et al., BioTechniques 63 (2017), 205-214). Further, the authors need to show which other HCMV proteins might be involved in downregulating FcRn.*

Responses: Our co-author Dr. Jeffrey Cohen communicated with Gerhard Jahn the corresponding author (Schempp S et al., Virus Research 155 (2011), 446-454) to request US11 knockout virus. Unfortunately, we did not receive a response after several attempts. It would take many months to make a US11 knockout virus from the clinical HCMV strain. We reason the either US11 transfection or US11 specific siRNA knockdown has clearly demonstrated US11 specifically mediates FcRn degradation in either transfected or virally-infected cells.

Our parallel studies showed that HCMV [redacted] protein also bound to FcRn (Fig. 1); however, we did not find evidence either [redacted] protein mediates FcRn degradation. These additional HCMV proteins would affect FcRn function and make the experiments more complexed in studying the impact of US11 on FcRn functions in US11 mutant virus-infected cells (Page 19, bottom). We are currently performing studies to detail how [redacted] protein blocks FcRn function. A new manuscript will publish these results and cite this submitted manuscript as a reference. Hence, we prefer to not disclose the detailed information about the interaction of FcRn with either [redacted] protein (see the response to reviewer 1, comment 1). In addition, this US11 manuscript already contains large volume of information and many Figures.

9. What are the implications for primary infection and/or reactivation in pregnant women, if any? Is there a role for FcRn in superinfection (secondary HCMV infection) when a pregnant woman has already been infected by CMV?

Responses: Reviewer raises an intriguing question. Mothers who are CMV seropositive prior to pregnancy can also develop a secondary CMV infection either due to reactivation of virus residing at specific sites in the body or reinfection with a different viral strain. The neutralizing antibody is important against congenital HCMV infection, however, the results of administration of CMV hyperimmune globulin to acutely CMV-infected pregnant women to protect their fetuses have been mixed. We reason that FcRn degradation by US11 would reduce placental transport of neutralizing IgG and thwart antibody-mediated protection. However, it was also reported that the preexisting, nonneutralizing maternal IgG has been implicated in facilitation of placental transmission of IgG-HCMV complexes can, especially during the third trimester when the placental IgG transfer peaks. The study in this manuscript represents the first study to substantially show HCMV or US11 mediates FcRn degradation. This study will help us to further understand assess the role of maternal antibody function and placental transmission in an ex vivo human placental model.

Minor points.

Page 4, introduction, second paragraph: HCMV has been extremely successful in infecting humans...". While it is true that CMV is a master of immune evasion, many herpesviruses have been successful in infecting humans and seroprevalence is higher among, e.g., HSV and VZV. With HCMV, there are geographical regions where only approximately 50% of the population are infected. The beginning of the sentence might be reworded.

Responses: We agree with the reviewer, we have rephrased the sentence so that it reads (page 3, second paragraph of the revised paper). "HCMV has been successful in infecting humans due to its ability to evade the immune system and establish lifelong latency and persistent virus shedding".

Page 4, line 8 from the bottom: ...the US3 protein...

Responses: We have corrected this error on page 3, second paragraph of the revised paper.

Page 6, bottom paragraph: Human CD34-positive cells (hematopoietic stem cells) should be included in the list of cell types that are infected by HCMV. Is FcRn expressed at similar levels in these cell types?

Responses: The reviewer raised an excellent point, we have added this information into the revised manuscript (page 5, first full paragraph of the revised paper). A new reference is cited in the revised manuscript.

45. Maciejewski JP, Bruening EE, Donahue RE, Mocarski ES, Young NS, St Jeor SC. Infection of hematopoietic progenitor cells by human cytomegalovirus. *Blood*. 1992 Jul 1;80(1):170-8.

Page 9, bottom paragraph: The introductory sentence is incomplete.

Responses: We have rephrased the sentence.

Page 43, figure legends. First line: The sentence is incomplete.

Responses: Thanks for pointing out his issue. We have deleted this incomplete sentence in the Figure 1 legend.

Page 52, figure 9. What determines the portion of beta2-microglobulin-free FcRn HC molecules that associate with US11 in the ER?

Responses: The β 2-microglobulin is complexed with MHC class I and other MHC class I-related molecules, such as HFE, CD1, FcRn, HLA-G, etc. Therefore, the portion of β 2-microglobulin-free FcRn HC molecules that associate with US11 in the ER will be determined by the expression levels of β 2-microglobulin, FcRn, and US11.

REVIEWERS' COMMENTS:

Reviewer #1 (Remarks to the Author):

The authors have now sufficiently addressed my concerns and the manuscript is suitable for publication.

Reviewer #2 (Remarks to the Author):

The revised manuscript by Liu et al describes the novel finding that HCMV US11 targets the cellular protein neonatal Fc receptor (FcRN) for degradation in a proteasomal dependent manner. The current form of the manuscript describes these findings in well-controlled experiments that support the author's conclusions. They have satisfactorily addressed this reviewer's comments through modifying the text and including additional experiments.

Reviewer #3 (Remarks to the Author):

In their revised manuscript, the authors have answered many of the concerns raised by the reviewers and provide new useful data to strengthen their findings. However, a couple of items remain.

1. In response to question 1 of reviewer 1, it is stated that the clinical isolate "has not been completely sequenced". I would assume that at least US11 has been sequenced and the sequence should be made available in GenBank.

2. Data shown in the rebuttal letter should be included in the manuscript.

3. The finding that HCMV downregulates cell surface or intracellular FcRn in human foreskin fibroblasts published by Fielding et al. in eLife could not be reproduced by the authors. This needs to be discussed.

4. I agree that US11 mutation would take long if the clinical isolate has not been cloned as a BAC yet. However, a BAC is available (Laib Sampaio et al.) from which US11 could be easily deleted; in my opinion, employing a knockout virus in the experiments would strengthen the findings and would be helpful to evaluate the other CMV proteins in question.

REVIEWERS' COMMENTS:

Reviewer #1 (Remarks to the Author):

The authors have now sufficiently addressed my concerns and the manuscript is suitable for publication.

Responses: We appreciate the reviewer's constructive and positive comments.

Reviewer #2 (Remarks to the Author):

The revised manuscript by Liu et al describes the novel finding that HCMV US11 targets the cellular protein neonatal Fc receptor (FcRN) for degradation in a proteasomal dependent manner. The current form of the manuscript describes these findings in well-controlled experiments that support the author's conclusions. They have satisfactorily addressed this reviewer's comments through modifying the text and including additional experiments.

Responses: We appreciate the reviewer's constructive and positive comments.

Reviewer #3 (Remarks to the Author):

In their revised manuscript, the authors have answered many of the concerns raised by the reviewers and provide new useful data to strengthen their findings. However, a couple of items remain.

Responses: We thank the reviewer's positive comments.

1. In response to question 1 of reviewer 1, it is stated that the clinical isolate "has not been completely sequenced". I would assume that at least US11 has been sequenced and the sequence should be made available in GenBank.

Responses: The reviewer raised an excellent point. We agree with the reviewer, we have made the US11 sequence of the clinical isolate available in the Genbank with an accession number MK647994. We have added this information in the revised manuscript.

2. Data shown in the rebuttal letter should be included in the manuscript.

Responses: We included the majority of data in the rebuttal letter in the revised manuscript. To respect the Reviewer, we have also included FcRn expression in the fibroblast as a supplementary Figure.

Although we found [redacted] interacting with FcRn, our parallel studies did not find evidence that either [redacted] protein mediated FcRn degradation. We are currently per-

forming studies to detail how [redacted] protein blocks FcRn function. The new manuscripts will report these results and we will cite this submitted US11 manuscript as a reference. Since the making known of the [redacted] interacting with FcRn affects the submission of our new manuscripts, we prefer to not include the detailed information about the interaction of FcRn with either [redacted] protein in this manuscript. In addition, this US11-focusing manuscript already contains large volume of information and many Figures. We are sure the reviewer understands this point.

The Reviewer 1 also raised this question and additional discussion can be found in the response to Reviewer 1 Question 1. We have shown that HCMV [redacted], US11, and [redacted] interact with FcRn and that FcRn is degraded during HCMV infection. However, [redacted] and [redacted] did not cause FcRn degradation but it was observed with US11. Therefore US11 has an unique and distinct impact on FcRn. We are continuing to study mechanisms of action for [redacted] as they relate to FcRn biology, with NIH grant support. We hope to publish soon on [redacted], and to compare with results in the current manuscript.

In the paper published by Fielding et al. (2017), US20 region affected the expression levels of 25 surface proteins or 16 intracellular proteins, suggesting a non-specific regulation. It remains unclear how US20 protein affects different proteins, more detailed analysis or work needs to be done in the future. Hence, we prefer to not discuss or include US20 finding in our manuscript.

3. The finding that HCMV downregulates cell surface or intracellular FcRn in human foreskin fibroblasts published by Fielding et al. in eLife could not be reproduced by the authors. This needs to be discussed.

Responses: We agree with the reviewer to discuss this discrepancy in the revised manuscript, page 22. To do this, we have also included the detection of FcRn expression in the fibroblast in the Supplementary Figures.

Discussion: "HCMV downregulated FcRn expression in human foreskin fibroblast (HFF) which was detected by mass spectrometry (23). However, we failed to detect FcRn expression by Western blot analysis in either uninfected or HCMV infected HFF (Supplementary Fig. 12). This discrepancy may be caused by the protein detection method or the low level of FcRn expression in the HFF cells. More experiments are needed to verify this result (23)".

4. I agree that US11 mutation would take long if the

Fig 1. HCMV or HCMV Δ US1-12-infected HFF or HUVEC cells. HFF or HUVEC cells were grown on glass coverslips and infected with virus at an MOI of 5. At day 2 p.i., monolayers were fixed with 4% paraformaldehyde and permeabilized in 0.2% Triton X-100. Subsequently, the cells were incubated with affinity-purified anti-pp65 (Green) specific Ab, followed by Alexa Fluoro 555-conjugated IgG. Staining that appears yellow in the merged images indicates colocalization of US11 with pp65. The nuclei were stained with DAPI

clinical isolate has not been cloned as a BAC yet. However, a BAC is available (Laib Sampaio et al.) from which US11 could be easily deleted; in my opinion, employing a knockout virus in the experiments would strengthen the findings and would be helpful to evaluate the other CMV proteins in question.

Responses: We contacted Dr. Kerstin Laib Sampaio in Germany about TB40-BAC-KL7-SE-EGFP clone. Dr. Sampaio mentioned that this HCMV mutant is disrupted both US11 and the US2-6 region. In addition, the UL40 ORF was also repaired on this background. Dr. Sampaio recommended us to contact Dr. Anne Halenius in Freiburg who might generated a single US11 deletion on the basis of TB40-BAC-KL7-SE-EGFP. Dr. Halenius responded that there is no single US11 deletion in her lab; her lab constructed a TB40E BAC4 mutant lacking the genes US2-6 and US11 as well. It would take many months to make a US11 knockout virus from the clinical HCMV strain if we are lucky.

Although we emphasized that the either US11 transfection or US11 specific siRNA knockdown has demonstrated that US11 specifically mediates FcRn degradation in either transfected or virally-infected cells. To respect the Reviewer 3' comment, our co-author Dr. Cohen's lab has a mutant HCMV Δ US1-12 lacking the US1-12 genes. HCMV Δ US1-US12 (80) was a kind gift from Dr. Fenyong Liu, University of California, Berkeley and Dr. Hua Zhu, New Jersey Medical School, Rutgers University. We tested this HCMV Δ US1-12 strain, it infected the fibroblast HFF cell lines, however, HCMV Δ US1-12 was not capable of infecting primary endothelial cell HUVEC in comparison with that of HCMV Bethesda BAL strain (Fig 1).

To show the possibility that the HCMV Δ US1-12 fails to degrade FcRn, we first established a stable HFF cell line expressing FcRn receptor. To further characterize the HCMV Δ US1-12, we infected HFF^{FcRn} with clinical HCMV or HCMV Δ US1-12 (MOI 5) (Fig 2). The infection was verified by detecting HCMV phosphoprotein 65 (PP65). At day 2 post-infection (p.i.), we determined that anti-US11 Ab co-immunoprecipitated with FcRn in HCMV-infected HFF^{FcRn} cells (Fig. 2, lane 1), but not pulled down FcRn in HCMV Δ US1-12-infected HFF^{FcRn} cells. This result confirmed that HCMV Δ US1-12 indeed lacks US11 gene expression.

To specifically examine the rate of FcRn HC degradation, 48 hr post-infection, we performed a quantitative cycloheximide (CHX) chase assay. HFF^{FcRn} cells were treated with CHX (100 μ g/ml) and FcRn intensity detected in Western blot was measured by an NIH Imager for the indicated times. In HFF^{FcRn} cells, the HCMV infection induced a significant and time-dependent decrease in FcRn expression (Fig. 3A+3D) in comparison with that of HCMV Δ US1-12-infected HFF^{FcRn} cells (Fig. 3B+3D). FcRn showed the long-term stability in mock-infected

Fig 2. HCMV Δ US1-12 mutant virus does not express US11 protein. HFF^{FcRn} cells were infected with HCMV or HCMV Δ US1-12 virus at an MOI of 5. The infection was verified by detecting HCMV phosphoprotein 65 (PP65). At day 2 p.i., the cell lysates from infected cells were immunoprecipitated with anti-US11 Ab. The immunoprecipitates were subjected to 12% SDS-PAGE electrophoresis under reducing conditions, then transferred to a nitrocellulose membrane for Western blotting with anti-FcRn or US11 Ab as indicated. The cell lysates (20 mg) were blotted as controls. Immunoblots (IB) were developed with ECL.

cells (Fig. 3C+3D). The β_2m levels were unaffected in HFF^{FcRn} cells (Fig. 3E). Therefore, the HCMV Δ US1-12 strain fails to promote FcRn protein degradation.

We have to admit that several genes in the HCMV Δ US1-12 are impaired simultaneously, we are not able to draw a solid conclusion that the absence of US11 expression is responsible for the failure of FcRn degradation in HCMV Δ US1-12-infected HFF^{FcRn} cells. We add this information in the revised manuscript, page 10.

“In addition, we also found a mutant HCMV Δ US1-12 virus that lacks the US1-12 genes failed to induce a significant and time-dependent decrease in FcRn expression in HCMV Δ US1-12-infected HFF^{FcRn} cells in comparison with that of mock of HCMV-infected cells (Supplementary Fig. 7)”.

Fig 3. HFF^{FcRn} cells were infected with clinic strain HCMV (MOI 5) (A), HCMV Δ US1-12 (B), or mock-infected (C) for 48 hr. 48 hr later, cells were then treated with CHX (100 μ g/ml) for the indicated time. The cells were lysed after CHX treatment, protein levels were measured, and Western blotting and ECL were performed. The level of remaining endogenous FcRn (D) and β_2m (E) at different time points was quantified as the percentage of the β -tubulin level. The percentage of time point 0 (min) is assigned a value of 100% and the values from other time points are normalized to this value. Each experiment was carried out three times.